# Continual Offline Reinforcement Learning via Diffusion-based Dual Generative Replay

## Abstract

We study continual offline reinforcement learning, a practical paradigm that facilitates forward transfer and mitigates catastrophic forgetting to tackle sequential offline tasks. We propose a dual generative replay framework that retains previous knowledge by concurrent replay of generated pseudo-data. First, we decouple the continual learning policy into a diffusion-based generative behavior model and a multi-head action evaluation model, allowing the policy to inherit distributional expressivity for encompassing a progressive range of diverse behaviors. Second, we train a task-conditioned diffusion model to mimic state distributions of past tasks. Generated states are paired with corresponding responses from the behavior generator to represent old tasks with high-fidelity replayed samples. Finally, by interleaving pseudo samples with real ones of the new task, we continually update the state and behavior generators to model progressively diverse behaviors, and regularize the multi-head critic in a behavior cloning manner to mitigate forgetting. Experiments on various benchmarks demonstrate that our method achieves better forward transfer with less forgetting, and closely approximates results of using previous ground-truth data due to its high-fidelity replay of the sample space.

## 1 Introduction

Offline reinforcement learning (RL) (Fujimoto et al., 2019; Levine et al., 2020) allows an agent to learn from a pre-collected dataset without having to interact with the environment in real time. This learning paradigm is vital for many realistic scenarios where collecting data online can be very expensive or dangerous, such as robotics (Kumar et al., 2022), autonomous driving (Yu et al., 2018), and healthcare (Gottesman et al., 2019), and has attracted widespread attention in recent years (Yuan & Lu, 2022; Nikulin et al., 2023). The emergence of offline RL also holds tremendous promise for turning massive datasets into powerful sequential decision-making engines, e.g., decision transformers (Chen et al., 2021), akin to the rise of large language models like GPT (Brown et al., 2020).

In the real world, huge amounts of new data are produced as new tasks emerge overwhelmingly (van de Ven et al., 2022). However, current parametric RL models learn representations from stationary batches of training data, and are prone to forgetting previously acquired knowledge when tackling new tasks, a phenomenon known as catastrophic forgetting or interference (Parisi et al., 2019). Accordingly, continual RL (Khetarpal et al., 2022), also known as lifelong RL, is widely studied to address two major issues: i) mitigating catastrophic forgetting, and ii) allowing forward transfer, i.e., leveraging previous knowledge for efficient learning of new tasks. In recent years, continual learning has seen the proposal of a variety of methods (Fu et al., 2022; Gaya et al., 2023) that can mainly be categorized into: regularization-based (Zeng et al., 2019; Kaplanis et al., 2019), parameter isolation (Kessler et al., 2022; Konishi et al., 2023), and rehearsal methods Isele & Cosgun (2018); Rolnick et al. (2019); Daniels et al. (2022). Among them, rehearsal with experience replay is a popular choice due to its simplicity and promising results (Gao & Liu, 2023). In a recent benchmark study Wolczyk et al. (2022), experiments demonstrate that experience replay significantly improves both transfer and the final performance, and plays a critical role in continual RL.

This paper focuses on continual offline RL (CORL), an understudied problem that lies in the intersection of offline RL and continual RL. The learner is presented with a sequence of offline tasks where datasets are collected by different behavior policies. Current rehearsal-based methods are confronted with two main challenges arising from intrinsic attributes of offline RL. First, differ-

ent from supervised learning, RL models are more prone to deficient generalization across diverse tasks (Kirk et al., 2023), and existing policy models are usually unimodal Gaussian models with limited distributional expressivity (Chen et al., 2023). Nonetheless, in the realm of CORL, collected behaviors become progressively diverse as novel datasets continue to emerge, which might lead to performance degradation due to deficient generalization and distributional discrepancy. Second, existing methods (Rolnick et al., 2019; Wolczyk et al., 2022; Gai et al., 2023) rely on a substantial buffer for storing real samples from previous tasks. However, this presents a memory capacity constraint that becomes more pronounced as new tasks keep emerging, restricting its applicability for large-scale problems as well as practical scenarios involving privacy issues.

To address the above challenges, we propose an efficient **C**ontinual learning method via diffusion-based d**u**al **G**enerative **R**eplay for **O**ffline RL (CuGRO), which avoids storing past samples and retains previous knowledge by concurrent replay of generated pseudo-data. First, inspired by Chen et al. (2023), we decouple the continual learning policy into an expressive generative behavior model $\mu_{\boldsymbol{\phi}}(\boldsymbol{a}|\boldsymbol{s})$ and an action evaluation model $Q_{\boldsymbol{\theta}}(\boldsymbol{s}, \boldsymbol{a})$. Training a unified behavior model can continually absorb new behavior patterns to promote forward knowledge transfer for the offline setting, and sampling from this generative model can naturally encompass a progressive range of observed behaviors. Second, we introduce a state generative model $p_{\boldsymbol{\varphi}}(\boldsymbol{s}|k)$ to mimic previous state distributions conditioned on task identity $k$. Generated states $\hat{\boldsymbol{s}} \sim p_{\boldsymbol{\varphi}}(\boldsymbol{s}|k)$ are paired with corresponding responses from the behavior generative model $\hat{\boldsymbol{a}} \sim \mu_{\boldsymbol{\phi}}(\boldsymbol{a}|\hat{\boldsymbol{s}})$ to represent old tasks. In particular, we leverage existing advances in diffusion probabilistic models (Ho et al., 2020) to model states and corresponding behaviors with high fidelity, allowing the continual policy to inherit the distributional expressivity. Finally, to model progressively diverse behaviors without forgetting, we interleave replayed samples with real ones of the new task to continually update the state and behavior generators. We use a multi-head critic to tackle the diversity of emerging tasks, and mitigate forgetting of the critic in a behavior cloning manner. In summary, our main contributions are threefold:

- We propose an efficient generative replay framework for CORL. To the best of our knowledge, CuGRO is the first that leverages expressive diffusion models to tackle the CORL challenge.

- We develop a dual generator system to synthesize high-fidelity samples for modeling progressively diverse behaviors, and mitigate forgetting of a multi-head critic using behavior cloning.

- We empirically show on various benchmarks that CuGRO better mitigates forgetting and facilitates forward transfer than prior methods, and closely approximates the same results as using previous ground-truth data due to its reliable and high-fidelity synthesis of the sample space.

## 2 PRELIMINARIES

### 2.1 OFFLINE REINFORCEMENT LEARNING

RL is commonly studied based on the Markov decision process (MDP) formulation, $(\mathcal{S}, \mathcal{A}, P, r, \gamma)$, where $\mathcal{S}$ and $\mathcal{A}$ denote the state and action spaces, $P(\boldsymbol{s}'|\boldsymbol{s}, \boldsymbol{a})$ and $r(\boldsymbol{s}, \boldsymbol{a})$ are the transition and reward functions, and $\gamma$ is the discount factor. The goal is to maximize the expected return:

$$J(\pi) = \mathbb{E}_{\boldsymbol{s} \sim \rho_{\pi}(\boldsymbol{s})} \mathbb{E}_{\boldsymbol{a} \sim \pi(\cdot|\boldsymbol{s})} \left[ \sum r(\boldsymbol{s}, \boldsymbol{a}) \right] = \int_{\mathcal{S}} \rho_{\pi}(\boldsymbol{s}) \int_{\mathcal{A}} \pi(\boldsymbol{a}|\boldsymbol{s}) Q^{\pi}(\boldsymbol{s}, \boldsymbol{a}) \mathrm{d}\boldsymbol{a} \mathrm{d}\boldsymbol{s}, \quad (1)$$

where $\rho_{\pi}(\boldsymbol{s})$ is the state visitation frequencies induced by policy $\pi$. When online data collection from $\pi$ is feasible, it is difficult to estimate $\rho_{\pi}(\boldsymbol{s})$ and thus $J(\pi)$. For a static dataset $\mathcal{D}_{\mu} = \sum_i (\boldsymbol{s}_i, \boldsymbol{a}_i, r_i, \boldsymbol{s}'_i)$ collected by a behavior policy $\mu(\boldsymbol{a}|\boldsymbol{s})$, offline RL approaches (Peng et al., 2019; Nair et al., 2020) usually encourage $\pi$ to stick with $\mu$ and maximize a constrained objective as

$$J'(\pi) = \int_{\mathcal{S}} \rho_{\mu}(\boldsymbol{s}) \int_{\mathcal{A}} \pi(\boldsymbol{a}|\boldsymbol{s}) Q^{\pi}(\boldsymbol{s}, \boldsymbol{a}) \mathrm{d}\boldsymbol{a} \mathrm{d}\boldsymbol{s} - \frac{1}{\alpha} \int_{\mathcal{S}} \rho_{\mu}(\boldsymbol{s}) D_{\mathrm{KL}} \left( \pi(\cdot|\boldsymbol{s}) || \mu(\cdot|\boldsymbol{s}) \right) \mathrm{d}\boldsymbol{s}. \quad (2)$$

The optimal policy $\pi^*$ for Eq. (2) can be derived by using a Lagrange multiplier as

$$\pi^*(\boldsymbol{a}|\boldsymbol{s}) = \frac{1}{Z(\boldsymbol{s})} \mu(\boldsymbol{a}|\boldsymbol{s}) \exp\left( \alpha Q^*(\boldsymbol{s}, \boldsymbol{a}) \right), \quad (3)$$

where $Z(\boldsymbol{s})$ is the partition function. By projecting $\pi^*$ onto a parameterized policy $\pi_{\boldsymbol{\phi}}$ with a critic $Q_{\boldsymbol{\theta}}$, we can obtain the final objective in a weighted regression form as

$$\arg\min_{\boldsymbol{\phi}} \mathbb{E}_{\boldsymbol{s} \sim \mathcal{D}_{\mu}} [D_{\mathrm{KL}} (\pi^* || \pi_{\boldsymbol{\phi}})] = \arg\max_{\boldsymbol{\phi}} \mathbb{E}_{(\boldsymbol{s}, \boldsymbol{a}) \sim \mathcal{D}_{\mu}} \left[ \frac{1}{Z(\boldsymbol{s})} \log \pi_{\boldsymbol{\phi}}(\boldsymbol{a}|\boldsymbol{s}) \exp\left( \alpha Q_{\boldsymbol{\theta}}(\boldsymbol{s}, \boldsymbol{a}) \right) \right]. \quad (4)$$

## 2.2 Generative Behavior Modeling

To avoid learning an explicitly parameterized policy model in Eq. (4), Chen et al. (2023) decouples the learned policy into an expressive generative behavior model $\mu_\phi(\boldsymbol{a}|\boldsymbol{s})$ and an action evaluation model $Q_{\boldsymbol{\theta}}(\boldsymbol{s}, \boldsymbol{a})$, and form a policy improvement step as

$$\pi(\boldsymbol{a}|\boldsymbol{s}) \propto \mu_\phi(\boldsymbol{a}|\boldsymbol{s}) \cdot \exp\left(\alpha Q_{\boldsymbol{\theta}}(\boldsymbol{s}, \boldsymbol{a})\right), \tag{5}$$

where $\alpha$ is a temperature parameter that balances between conservative and greedy improvements. Diffusion probabilistic models (Ho et al., 2020; Song et al., 2021) are utilized to fit the behavior distribution from the offline dataset. A state-conditioned diffusion model $\boldsymbol{\epsilon}_\phi$ is trained to predict the noise $\boldsymbol{\epsilon}$ added to the action $\boldsymbol{a}$ sampled from the behavior policy $\mu(\cdot|\boldsymbol{s})$ as

$$\arg\min_{\phi} \mathbb{E}_{(\boldsymbol{s}, \boldsymbol{a}) \sim D_\mu, \boldsymbol{\epsilon}, t}\left[\left\|\sigma_t \boldsymbol{\epsilon}_\phi\left(\alpha_t \boldsymbol{a} + \sigma_t \boldsymbol{\epsilon}, \boldsymbol{s}, t\right) + \boldsymbol{\epsilon}\right\|_2^2\right] \tag{6}$$

where $t \sim \mathcal{U}(0, T)$, $\boldsymbol{\epsilon} \sim \mathcal{N}(0, \boldsymbol{I})$, and $\alpha_t$ and $\sigma_t$ are determined by the forward diffusion process. The generative model $\boldsymbol{\epsilon}_\phi$ is trained to denoise the perturbed action $\boldsymbol{a}_t := \alpha_t \boldsymbol{a} + \sigma_t \boldsymbol{\epsilon}$ back to the original undisturbed one $\boldsymbol{a}$. Then, a critic network $Q_{\boldsymbol{\theta}}(\boldsymbol{s}, \boldsymbol{a})$ is learned to evaluate actions sampled from the learned behavior model $\mu_\phi(\cdot|\boldsymbol{s})$. With the idea of episodic learning (Blundell et al., 2016; Ma et al., 2022), the value function is updated using a planning-based Bellman operator as

$$R_i^{(j)} = r_i + \gamma \max\left(R_{i+1}^{(j)}, V_{i+1}^{(j-1)}\right),$$
$$\text{where} \quad V_i^{(j-1)} := \mathbb{E}_{\boldsymbol{a} \sim \pi(\cdot|\boldsymbol{s}_i)}\left[Q_{\boldsymbol{\theta}}(\boldsymbol{s}_i, \boldsymbol{a})\right], \tag{7}$$
$$\text{and} \quad \boldsymbol{\theta} = \arg\min_{\boldsymbol{\theta}} \mathbb{E}_{(\boldsymbol{s}_i, \boldsymbol{a}_i) \sim \mathcal{D}_\mu}\left[\left\|Q_{\boldsymbol{\theta}}(\boldsymbol{s}_i, \boldsymbol{a}_i) - R_i^{(j-1)}\right\|_2^2\right],$$

where $j = 1, 2, ...$ is the iteration number. Eq. (7) offers an implicit planning scheme within dataset trajectories to avoid bootstrapping over unseen actions and to accelerate convergence. Finally, the action is re-sampled using importance weighting with $\exp(\alpha Q_{\boldsymbol{\theta}}(\boldsymbol{s}, \boldsymbol{a}))$ being the sampling weights.

**Notation**. There are three "times" at play in this work: that of the RL problem, that of the diffusion process, and that of the continual learning setting. We use subscripts $i$ to denote the timestep within an MDP, superscripts $t$ to denote the diffusion timestep, and subscripts $k$ to denote the sequential task identity. For example, $\boldsymbol{a}_i^0$ refers to the $i$-th action in a noiseless trajectory, and $\boldsymbol{s}_k$ and $\phi_k$ represent a given state and the behavior model during task $M_k$.

## 3 Diffusion-based Dual Generative Replay

### 3.1 Problem Formulation and Method Overview

In CORL, we assume that the task follows a distribution $M_k = (\mathcal{S}, \mathcal{A}, P, r, \gamma)_k \sim P(M)$. The learner is presented with an infinite sequence of tasks $[M_1, ..., M_k, ...]$, and for each task $M_k$, an offline dataset $\mathcal{D}_{\mu_k} = \sum_i (\boldsymbol{s}_i, \boldsymbol{a}_i, r_i, \boldsymbol{s}_i')_k$ is collected by a behavior policy $\mu_k$. The learner can only access the offline dataset of the new task $M_K$, without retrieving real samples from previous tasks $\{M_k\}_{k=1}^{K-1}$ or any online environmental interactions. The objective for CORL is to learn a continual policy that maximizes the expected return over all encountered tasks as

$$J(\pi_{\text{continual}}) = \sum_{k=1}^{K} J_{M_k}(\pi_{M_k}). \tag{8}$$

Importantly, the learner ought to build upon the accumulated knowledge from previous tasks $M_1, ..., M_{K-1}$ to facilitate learning the new task $M_K$, while not forgetting previous knowledge.

In the following, we first introduce the dual generator system that mimics the state and behavior distributions of past tasks, as described in Section 3.2. Then, in Section 3.3, we present the dual generative replay mechanism that sequentially trains conditional diffusion-based generators to synthesize high-fidelity pseudo samples for modeling progressively diverse behaviors without forgetting. Finally, we show the behavior cloning process that prevents forgetting of a multi-head critic in Section 3.4. With the above implementations, the training procedure of CuGRO is illustrated in Fig. 1, and the algorithm summary is presented in Appendix A.

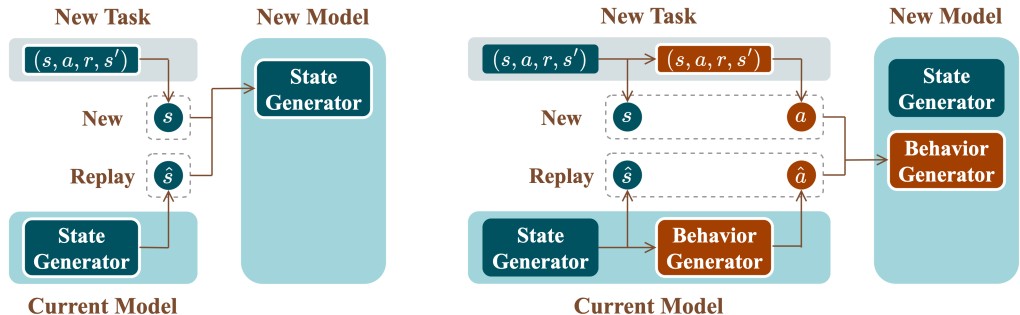

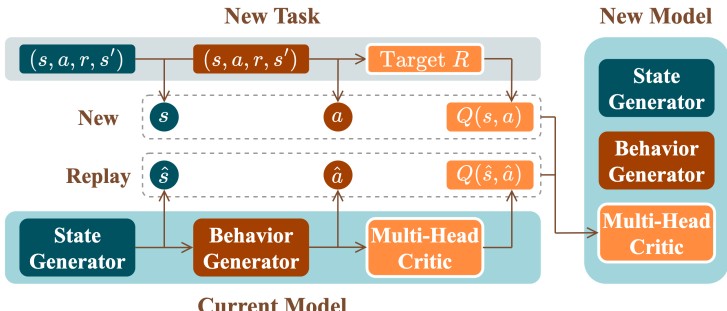

(c) Training the multi-head critic.

Figure 1: Sequential training of CuGRO. (a) A new diffusion-based state generative model is trained to mimic a mixed data distribution of real samples $s$ and replayed ones $\hat{s}$. (b) A new diffusion-based behavior generative model learns from real state-action pairs $(s, a)$ and pseudo pairs $(\hat{s}, \hat{a})$, where replayed action $\hat{a}$ is obtained by feeding replayed states $\hat{s}$ into current behavior generator. (c) A new head in the critic is expanded for tackling the new task with real state-action pairs and Bellman targets, and previous heads are regularized by cloning the Q-value of replayed pairs $(\hat{s}, \hat{a})$.

## 3.2 DUAL GENERATOR SYSTEM

**Behavior Generative Model**. Inspired by Chen et al. (2023), we decouple the continual learning policy into a diffusion-based generative behavior model $\mu_\phi(a|s)$ and an action evaluation model $Q_\theta(s, a)$ as in Eq. (5). This perspective rooted in generative modeling presents three promising advantages for CORL. First, existing policy models are usually unimodal Gaussians with limited distributional expressivity, while collected behaviors in CORL become progressively diverse as novel datasets keep emerging. [1] Learning a generative behavior model is considerably simpler since sampling from the behavior policy can naturally encompass a diverse range of observed behaviors, and allows the policy to inherit the distributional expressivity of diffusion models. Second, RL models are more prone to deficient generalization across diverse tasks (Kirk et al., 2023). [2] Learning a unified behavior model can naturally absorb novel behavior patterns, continually promoting knowledge transfer and generalization for offline RL. Third, generative behavior modeling can harness extensive offline datasets from a wide range of tasks with pretraining, serving as a foundation model to facilitate finetuning for any downstream tasks. This aligns with the paradigm of large language models (Brown et al., 2020), and we reserve this promising avenue for future research.

---

[1]For instance, given a bidirectional navigation task where the agent can go to either of two opposite directions to get the final reward, fitting the policy with an unimodal Gaussian distribution leads to encompassing the low-density area situated between the two peaks (Chen et al., 2023).

[2]For example, considering two navigation tasks where the goals are in adverse directions, the single RL model ought to execute completely opposite decisions under the same states for the two tasks. By contrast, a supervised classification model just needs to derive two decision boundaries (Li & Hoiem, 2017) or a merged one (Shin et al., 2017) based on the augmented feature space shared by two tasks.

**State Generative Model**. Under the generative replay framework, we train another diffusion-based generator to mimic state distributions of previous tasks $\{s_k\}_{k=1}^{K-1}$. Specifically, we learn a scored-based task-conditioned model $\epsilon_{\varphi}$ to predict the noise $\epsilon$ added to the state $s$ as

$$\varphi = \arg\min_{\varphi} \mathbb{E}_{s \sim \mathcal{D}_{\mu_k}, \epsilon, t} \left[ \|\sigma_t \epsilon_{\varphi} \left( \alpha_t s + \sigma_t \epsilon, k, t \right) + \epsilon\|_2^2 \right], \tag{9}$$

where $t \sim \mathcal{U}(0, T)$, $\epsilon \sim \mathcal{N}(0, \mathbf{I})$, and $\alpha_t$ and $\sigma_t$ are hyperparameters of the forward diffusion process, and state $s$ is sampled from the $k$-th task's dataset $\mathcal{D}_{\mu_k}$. The generative model $\epsilon_{\varphi}$ is trained to refine the perturbed state $s^t := \alpha_t s + \sigma_t \epsilon$ back to the original undisturbed one $s$, such that the random noise $s^T \sim \mathcal{N}(0, \mathbf{I})$ can be reversed to model the original state distribution.

To generate high-fidelity state samples for each past task, we condition the diffusion model on task identity $k$ as $\hat{s} \sim p_{\varphi}(s|k)$. Moreover, conditional diffusion models can further improve the sample quality with classifier guidance that trades off diversity for fidelity using gradients from a trained classifier (Dhariwal & Nichol, 2021). Unfortunately, in the CORL setting, it might be infeasible to train a classifier with task identity as the label, since new tasks continually emerge and samples from prior tasks are unavailable. We leave this line of model improvement as future work, such as training with classifier-free guidance (Ho & Salimans, 2022).

### 3.3 HIGH-FIDELITY DUAL GENERATIVE REPLAY

We consider sequential training on dual generative models $\{(\varphi_k, \phi_k)\}_{k=1}^K$, where the $K$-th models $(\varphi_K, \phi_K)$ learn the new task $M_K$ and the knowledge of previous models $(\varphi_{K-1}, \phi_{K-1})$. This involves a dual procedure of training the conditional diffusion-based state and behavior generators using the mechanism of generative replay, as described below.

**State Generative Replay.** At new task $M_K$, to prevent forgetting of the state generative model, we use the previously learned generator $\varphi_{K-1}$ to generate synthetic state samples for retaining knowledge of prior tasks, i.e., $\hat{s}_k \sim p_{\varphi_{K-1}}(s|k)$. The new state generator $\varphi_K$ receives real state samples of the new task $s_K$ and replayed state samples of all previous tasks $\{\hat{s}_k\}_{k=1}^{K-1}$. Real and replayed samples are mixed at a ratio that depends on the desired importance of the new task compared to the older ones. The state generator learns to mimic a mixed data distribution of real samples and replayed ones from the previous generator, aiming to reconstruct the cumulative state space. Formally, the loss function of the $K$-th state generative model during training is given as

$$\begin{aligned} \mathcal{L}_{\text{train}}(\varphi_K) = \ & \beta \, \mathbb{E}_{s_K \sim \mathcal{D}_{\mu_K}, \epsilon, t} \left[ \|\sigma_t \epsilon_{\varphi_K} \left( \alpha_t s_K + \sigma_t \epsilon, K, t \right) + \epsilon\|_2^2 \right] \\ & + (1 - \beta) \sum_{k=1}^{K-1} \mathbb{E}_{\hat{s}_k \sim p_{\varphi_{K-1}}(s|k), \epsilon, t} \left[ \|\sigma_t \epsilon_{\varphi_K} \left( \alpha_t \hat{s}_k + \sigma_t \epsilon, k, t \right) + \epsilon\|_2^2 \right], \end{aligned} \tag{10}$$

where $\beta$ is the ratio of mixing real data. As we aim to evaluate the generator on original tasks, test loss differs from the training loss as

$$\begin{aligned} \mathcal{L}_{\text{test}}(\varphi_K) = \ & \beta \, \mathbb{E}_{s_K \sim \mathcal{D}_{\mu_K}, \epsilon, t} \left[ \|\sigma_t \epsilon_{\varphi_K} \left( \alpha_t s_K + \sigma_t \epsilon, K, t \right) + \epsilon\|_2^2 \right] \\ & + (1 - \beta) \sum_{k=1}^{K-1} \mathbb{E}_{s_k \sim \mathcal{D}_{\mu_k}, \epsilon, t} \left[ \|\sigma_t \epsilon_{\varphi_K} \left( \alpha_t s_k + \sigma_t \epsilon, k, t \right) + \epsilon\|_2^2 \right], \end{aligned} \tag{11}$$

**Behavior Generative Replay.** Subsequently, it is crucial to replay samples from previous tasks to continually train the behavior generative model $\mu_{\phi}$ for encompassing an expanding array of behavior patterns without forgetting. This update involves generating pseudo-state-action pairs using both the state and behavior generators. At the new task $M_K$, we first obtain synthetic state samples of previous tasks using learned state generator $\varphi_{K-1}$, and then pair these pseudo state samples with corresponding responses from the current behavior generator $\phi_{K-1}$ to represent old tasks as

$$\begin{aligned} \hat{s}_k &\sim p_{\varphi_{K-1}}(s|k), \\ \hat{a}_k &\sim \mu_{\phi_{K-1}}(a|\hat{s}_k), \quad k = 1, ..., K-1. \end{aligned} \tag{12}$$

The new behavior generator $\phi_K$ receives real state-action pairs of the new task $(s_K, a_K)$ and replayed pairs of all previous tasks $\{(\hat{s}_k, \hat{a}_k)\}_{k=1}^{K-1}$. The behavior generator aims to continually reconstruct the ever-expanding behavior patterns. With replayed pairs generated by Eq. (12), the loss

function of the $K$-th behavior generative model during training is given as

$$
\begin{aligned}
\mathcal{L}_{\text{train}}(\boldsymbol{\phi}_K) = \; & \beta \, \mathbb{E}_{(\boldsymbol{s}_K, \boldsymbol{a}_K) \sim \mathcal{D}_{\mu_K}, \boldsymbol{\epsilon}, t} \left[ \left\| \sigma_t \boldsymbol{\epsilon}_{\boldsymbol{\phi}_K} \left( \alpha_t \boldsymbol{a}_K + \sigma_t \boldsymbol{\epsilon}, \boldsymbol{s}_K, t \right) + \boldsymbol{\epsilon} \right\|_2^2 \right] \\
& + (1 - \beta) \sum_{k=1}^{K-1} \mathbb{E}_{(\hat{\boldsymbol{s}}_k, \hat{\boldsymbol{a}}_k), \boldsymbol{\epsilon}, t} \left[ \left\| \sigma_t \boldsymbol{\epsilon}_{\boldsymbol{\phi}_K} \left( \alpha_t \hat{\boldsymbol{a}}_k + \sigma_t \boldsymbol{\epsilon}, \hat{\boldsymbol{s}}_k, t \right) + \boldsymbol{\epsilon} \right\|_2^2 \right].
\end{aligned}
\tag{13}
$$

Similarly, we evaluate the behavior generator on original tasks, resulting in the test loss as

$$
\begin{aligned}
\mathcal{L}_{\text{test}}(\boldsymbol{\phi}_K) = \; & \beta \, \mathbb{E}_{(\boldsymbol{s}_K, \boldsymbol{a}_K) \sim \mathcal{D}_{\mu_K}, \boldsymbol{\epsilon}, t} \left[ \left\| \sigma_t \boldsymbol{\epsilon}_{\boldsymbol{\phi}_K} \left( \alpha_t \boldsymbol{a}_K + \sigma_t \boldsymbol{\epsilon}, \boldsymbol{s}_K, t \right) + \boldsymbol{\epsilon} \right\|_2^2 \right] \\
& + (1 - \beta) \sum_{k=1}^{K-1} \mathbb{E}_{(\boldsymbol{s}_k, \boldsymbol{a}_k) \sim \mathcal{D}_{\mu_k}, \boldsymbol{\epsilon}, t} \left[ \left\| \sigma_t \boldsymbol{\epsilon}_{\boldsymbol{\phi}_K} \left( \alpha_t \boldsymbol{a}_k + \sigma_t \boldsymbol{\epsilon}, \boldsymbol{s}_k, t \right) + \boldsymbol{\epsilon} \right\|_2^2 \right].
\end{aligned}
\tag{14}
$$

### 3.4 BEHAVIOR CLONING

To tackle the diversity of emerging tasks, we employ a multi-head critic network $\boldsymbol{\theta}$, with each head $Q_{\boldsymbol{\theta}}^k(\boldsymbol{s}, \boldsymbol{a})$ for capturing distinctive characteristics of each task $M_k$. When training the new task $M_K$, we use the planning-based operator in Eq. (7) to update the $K$-th Q-function $Q_{\boldsymbol{\theta}}^K(\boldsymbol{s}, \boldsymbol{a})$ with real samples from newest offline dataset $(\boldsymbol{s}_K, \boldsymbol{a}_K) \sim \mathcal{D}_{\mu_K}$. Then, we utilize a behavior cloning technique to alleviate forgetting of previous tasks' heads in the critic, analogous to the data rehearsal approaches that work well in supervised continual learning (Li & Hoiem, 2017; Chaudhry et al., 2019). After obtaining the pseudo state-action pairs of previous tasks $\{(\hat{\boldsymbol{s}}_k, \hat{\boldsymbol{a}}_k)\}_{k=1}^{K-1}$ in Eq. (12), we annotate these replayed samples using the previously trained critic $\boldsymbol{\theta}_{K-1}$ as $\{Q_{\boldsymbol{\theta}_{K-1}}^k(\hat{\boldsymbol{s}}_k, \hat{\boldsymbol{a}}_k)\}_{k=1}^{K-1}$. During training the new critic $\boldsymbol{\theta}_K$, we treat the labeled pseudo samples as expert data and perform behavior cloning via applying an auxiliary regularization term to imitate the expert data. The overall loss function for the multi-head critic is formulated as

$$
\begin{aligned}
\mathcal{L}(\boldsymbol{\theta}_K) = \; & \mathbb{E}_{(\boldsymbol{s}_K, \boldsymbol{a}_K) \sim \mathcal{D}_{\mu_K}} \left[ \| Q_{\boldsymbol{\theta}_K}^K(\boldsymbol{s}_K, \boldsymbol{a}_K) - R_K \|_2^2 \right] \\
& + \lambda \sum_{k=1}^{K-1} \mathbb{E}_{(\hat{\boldsymbol{s}}_k, \hat{\boldsymbol{a}}_k)} \left[ \left( Q_{\boldsymbol{\theta}_K}^k(\hat{\boldsymbol{s}}_k, \hat{\boldsymbol{a}}_k) - Q_{\boldsymbol{\theta}_{K-1}}^k(\hat{\boldsymbol{s}}_k, \hat{\boldsymbol{a}}_k) \right)^2 \right],
\end{aligned}
\tag{15}
$$

where $R_K$ corresponds to the Bellman target in Eq. (7), and $\lambda$ is the regularization coefficient. The first term is responsible for incorporating new knowledge using an expanded head in the critic, and the second term encourages the existing heads to stay close to previous outputs for mitigating forgetting. The critic, equipped with progressive heads, accumulates knowledge of previous tasks, enabling the reuse of the most relevant parts from the past to enhance the training of new tasks. Additionally, the behavior cloning loss acts as a regularizer that helps shape more general features, thus further improving forward knowledge transfer.

## 4 EXPERIMENTS

In this section, we show the applicability of our dual generative replay framework on various sequential learning tasks. Generative replay is superior to other continual learning methods in that the generator is the only constraint of the task performance. When the generative model is optimal, training the networks with generative replay is equivalent to joint training on the entire dataset. This is also a significant motivation for our method to leverage powerful diffusion models as our generators, as diffusion models have achieved sample quality superior to the current state-of-the-art generative models (Dhariwal & Nichol, 2021).

**Continual Offline Datasets.** We consider four domains from OpenAI Gym with the MuJoCo simulator (Todorov et al., 2012) as: 1) *Swimmer-Dir*, it trains a simulated swimmer to move towards a given 2D direction and tasks vary in the goal direction; 2) *Hopper-Vel*, it trains a simulated hopper to run forward at a particular velocity and tasks vary in the goal velocity; 3) *Walker2D-Params*, it trains a two-dimensional two-legged walker to move forward and tasks differ in the physical parameters of the simulated robot; 4) *HalfCheetah-Vel*, it trains a planar cheetah to run forward at a given velocity and tasks differ in the goal velocity. For each evaluation domain, a sequence of four tasks with varying dynamics is presented, and the offline dataset collection is given in Appendix B.

**Baselines.** We compare the performance of CORL trained with variants of replay methods, including: 1) *Oracle*, it serves as the upper bound with exact replay by assuming a perfect generator.

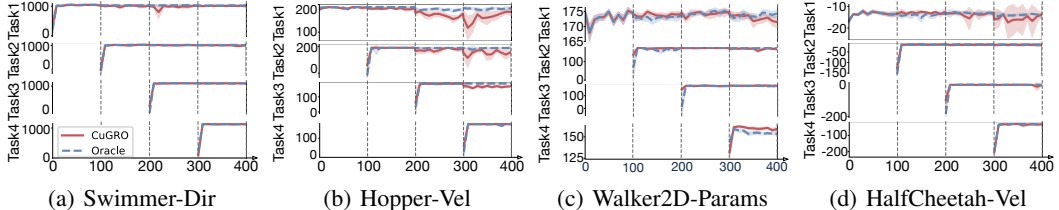

Figure 2: Test performance of CuGRO and Oracle on each task during sequential training implemented in all evaluation domains.

Therefore, Oracle uses real samples of all previous tasks to sequentially train the dual generators, and to train the multi-head critic with behavior cloning. 2) *Noise*, it considers the opposite case when generated samples do not resemble the real distribution at all. We omit the state generative model and use random noises as replayed state samples. The behavior generator and multi-head critic are sequentially trained using these randomly generated states. 3) *None*, it naively trains the model pipeline on sequential tasks without replay. We use the same notations throughout this section.

For evaluation, all experiments are carried out with 10 different random seeds, and the mean of the received return is plotted with 95% bootstrapped confidence intervals of the mean (shaded). The standard errors are presented for numerical results. Appendix C gives implementation details and hyperparameter settings of CuGRO.

## 4.1 MAIN RESULTS

Our primary concern is to validate whether the dual generative replay mechanism can realize high-fidelity replay of the sample space. Hence, we first look into the performance gap between CuGRO and Oracle to see whether our method can closely approximate the results of using previous ground-truth data. Fig. 2 presents the test performance on each task during sequential training. The continual RL policy with dual generative replay (red) maintains the former task performance throughout sequential training on multiple tasks, and obtains almost the same performance as the Oracle (blue) in most evaluation domains. Specially, in the Hopper-Vel domain, the performance of CuGRO for retaining previous knowledge drops a little in later task training, as shown in Fig. 2(b). It is likely because the Hopper robot easily falls down and terminates the episode during training, and switching the goal velocity during sequential training exacerbates this phenomenon. Hence, the continual policy becomes extremely sensitive to tiny extrapolation errors between replayed pseudo samples and the ground-truth data of previous tasks. However, in domains with higher dimensions like Walker2D-Params and HalfCheetah-Vel, CuGRO can still achieve almost full performance compared to real experience replay. It demonstrates CuGRO's capability of accurately modeling the mixed data distribution to produce reliable continual learning performance.

Furthermore, Fig. 3 shows the average test performance of CuGRO and baseline methods measured on cumulative tasks during sequential training, and Table 1 presents the corresponding final performance averaged over all sequential tasks. Higher return is achieved when the replayed pseudo samples better resemble the real data. It can be observed that the None baseline with sequential training the model pipeline naively incurs catastrophic forgetting (green). The average performance of None keeps decreasing significantly, indicating that the forgetting becomes severer as the sequential training goes on. Similarly, the Noise baseline that replays random Gaussian noises with recorded responses does not help the continual learning performance (purple). An interesting point is that Noise outperforms None by a large margin. It shows that adding random noises to the sample space could actually augment the feature space with enhanced generalization and robustness, thus helping alleviate forgetting to some extent.

## 4.2 HYPERPARAMETER ANALYSIS

In replay-based continual learning, the coefficient $\lambda$ in Eq. (15) is a key factor that balances the learning of the expanded new head (plasticity) and the behavior cloning of previous knowledge (stability). In the CuGRO setting, we select $\lambda$ as 1 and achieve good performance in all evaluation

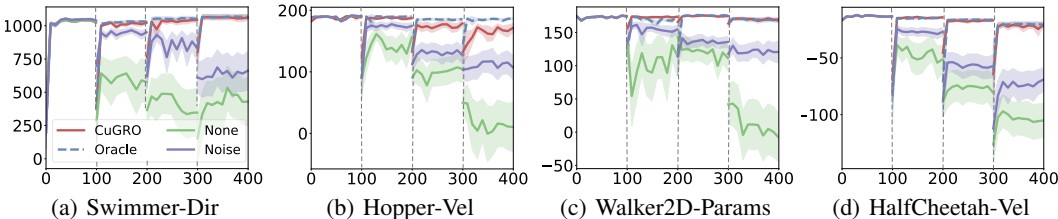

Figure 3: Average test performance of all methods over cumulative tasks during sequential training.

Table 1: Average final performance of all methods over all sequential tasks.

| Method | Swimmer-Dir | Hopper-Vel | Walker2D-Params | HalfCheetah-Vel |
|--------|-------------|------------|-----------------|-----------------|
| None | $429.50 \pm 569.74$ | $10.44 \pm 124.39$ | $-7.36 \pm 113.91$ | $-105.13 \pm 47.38$ |
| Noise | $661.26 \pm 447.12$ | $107.53 \pm 60.17$ | $120.61 \pm 51.51$ | $-69.22 \pm 32.94$ |
| Oracle | $\mathbf{1066.84 \pm 52.69}$ | $\mathbf{186.79 \pm 3.45}$ | $167.60 \pm 9.09$ | $-20.92 \pm 8.92$ |
| CuGRO | $1060.89 \pm 61.29$ | $171.13 \pm 22.15$ | $\mathbf{169.46 \pm 8.17}$ | $\mathbf{-20.89 \pm 6.29}$ |

domains. Moreover, we conduct experiments to analyze the influence of the tradeoff coefficient $\lambda$ on CuGRO's performance. Fig. 4 and Table 2 show the average performance over cumulative tasks and average final performance over all sequential tasks with varying coefficients, respectively. A higher coefficient corresponds to more emphasis on replaying previous tasks. Results demonstrate that CuGRO achieves good learning stability and robustness as the performance is insensitive to the pre-defined balancing coefficient. It verifies the effectiveness of the multi-head critic for tackling task diversity and the efficiency of behavior cloning for mitigating forgetting.

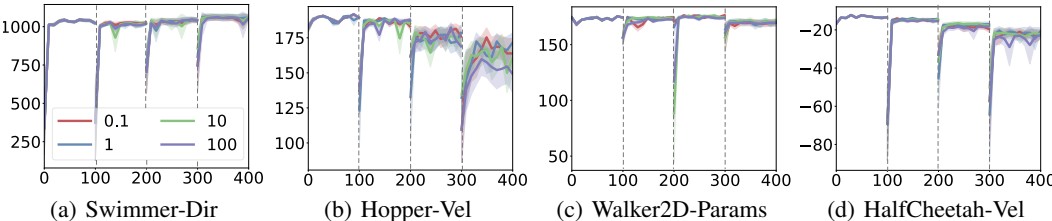

Figure 4: Average test performance of CuGRO over cumulative tasks with varying coefficients $\lambda$.

Table 2: Average final performance of CuGRO over all sequential tasks with varying coefficients $\lambda$.

| Method | Swimmer-Dir | Hopper-Vel | Walker2D-Params | HalfCheetah-Vel |
|--------|-------------|------------|-----------------|-----------------|
| 0.1 | $1055.41 \pm 61.24$ | $163.91 \pm 32.91$ | $169.56 \pm 8.94$ | $-22.34 \pm 7.15$ |
| 1 | $1060.89 \pm 61.29$ | $\mathbf{171.12 \pm 22.15}$ | $169.46 \pm 8.17$ | $\mathbf{-20.89 \pm 6.29}$ |
| 10 | $\mathbf{1062.79 \pm 59.35}$ | $160.16 \pm 39.00$ | $\mathbf{171.67 \pm 6.10}$ | $-22.43 \pm 6.73$ |
| 100 | $1053.36 \pm 60.66$ | $149.63 \pm 40.97$ | $169.74 \pm 6.25$ | $-22.81 \pm 6.59$ |

## 5 RELATED WORK

**Offline RL** imposes new challenges including overestimation of out-of-distribution (OOD) actions and accumulated extrapolation error due to distribution mismatch between the behavior and target policies (Levine et al., 2020). Policy regularization tackles the problem by constraining policy distribution discrepancy to avoid visiting state-action pairs less covered by the dataset (Kumar et al., 2019;

Fujimoto & Gu, 2021; Ran et al., 2023). Pessimistic value-based approaches learn a conservative Q-function for unseen actions to discourage the selection of OOD actions (Kostrikov et al., 2021b;a; Yang et al., 2022). An alternative is to learn a constrained actor with advantage-weighted regression to implicitly enforce a constraint on distribution shift and overly conservative updates (Peng et al., 2019; Nair et al., 2020; Wang et al., 2020). Recently, some studies (Kumar et al., 2019; Zhou et al., 2021; Chen et al., 2022) attempt to leverage advances in generative models, such as VAEs (Fujimoto et al., 2019) or diffusion models (Lu et al., 2023), to model offline datasets for model-based planning (Janner et al., 2022) or to directly model the policy with regularization (Wang et al., 2023).

**Continual RL** ought to facilitate forward transfer and mitigate catastrophic forgetting for sequential RL tasks. Regularization-based approaches imposes additional terms on the learning objective to penalize large updates on weights that are important for previous tasks, such as EWC (Kirkpatrick et al., 2017), OWM (Zeng et al., 2019), and PC (Kaplanis et al., 2019). With limited resources, comprising regularization terms might result in a compromise between achieving proficiency in both previous and new tasks. Parameter isolation approaches learn to mask a sub-network for each task in a shared network (Kang et al., 2022; Konishi et al., 2023), or dynamically expand model capacity to incorporate new information (Kessler et al., 2022; Wang et al., 2022), to prevent forgetting by keeping parameters of previous tasks unaltered. However, they could potentially suffer from over-consumption of network capacity as each task monopolizes and consumes some amount of resource.

Using the idea of episodic memory, rehearsal methods (Jeeveswaran et al., 2023) store examples from previous tasks that are reproduced for interleaving online updates when learning a new task (Isele & Cosgun, 2018; Rolnick et al., 2019; Wolczyk et al., 2022). Nonetheless, they might not be viable in real-world scenarios due to requiring a large working memory and involving several ramifications like data privacy or security concerns. To bypass the memory burden, generative replay methods (Shin et al., 2017) turn to use generative models to mimic older parts of the data distribution and sample synthetic data for rehearsal when learning the new task (Qi et al., 2023). Classical generative replay approaches typically adopt VAEs (Ketz et al., 2019) or GANs (Zhai et al., 2019) as the generator, and recent work suggests that powerful diffusion models can be employed with high-quality image synthesis (Gao & Liu, 2023). Ketz et al. (2019) continually trains a world model through the interleaving of generated episodes of past experiences. Daniels et al. (2022) uses a VAE generator to realize the replay of observation-action pairs in Starcraft II and Minigrid domains.

**Diffusion models** (Sohl-Dickstein et al., 2015) achieve great success in generating high-fidelity synthetic images (Dhariwal & Nichol, 2021), audios (Kim et al., 2022), and videos (Ho et al., 2022). The unconditional diffusion model does not require additional signals and learns the reverse process by estimating the noise at each step, such as DDPM (Ho et al., 2020) and its improvement (Nichol & Dhariwal, 2021). To accelerate sampling, DDIM (Song et al., 2020) generalizes the Markovian forward process of DDPM to a class of non-Markovian ones that lead to the same training objective. The conditional diffusion model relies on various source signals, e.g., class labels in image tasks, to generate corresponding data. Dhariwal & Nichol (2021) conditions on the gradient from a pretrained classifier to trade off diversity for fidelity, producing high-quality samples but not covering the whole distribution. Further, Ho & Salimans (2022) proposes classifier-free guidance that jointly trains a conditional and an unconditional diffusion model, and combines the resulting score estimates to attain a trade-off between sample quality and diversity. In this paper, we build our method on top of the diffusion model in (Song et al., 2021) that enables new sampling methods and further extends capabilities of score-based generative models through the lens of stochastic differential equations.

## 6 CONCLUSION AND DISCUSSION

In this paper, we tackle the CORL challenge via a dual generative replay framework that leverages advances in diffusion models to model states and corresponding behaviors with high fidelity, allowing the continual learning policy to inherit the distributional expressivity. The dual state and behavior generators are continually trained to model a progressive range of diverse behaviors via mimicking a mixed data distribution of real samples and replayed ones from previous generators. Experiments on various offline RL tasks verify the superiority of our method and its high-fidelity replay of the sample space. Though, our method requires two diffusion models to synthesize replayed samples, which could be further improved by sampling acceleration methods or developing one diffusion model for unifying the state and behavior modeling. We leave these directions as future work.

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
