## APPENDIX A. ALGORITHM SUMMARY

Based on the implementations in Section 3, we summarize the brief procedure of our method in Algorithm 1.

---

**Algorithm 1:** Continual offline RL via diffusion-based dual generative replay

---

**Input:** $(M_1, ..., M_K, ...)$: sequential tasks; $\quad \mathcal{D}_{\mu_k}$: offline dataset of task $M_k$, $k = 1, ..., K$;
$\qquad \mu_\phi(\boldsymbol{a}|\boldsymbol{s})$: state-conditioned behavior generative model with parameters $\phi$;
$\qquad p_\varphi(\boldsymbol{s}|k)$: task-conditioned state generative model with parameters $\varphi$;
$\qquad Q_\theta^k(\boldsymbol{s}, \boldsymbol{a})$: multi-head critic with parameters $\boldsymbol{\theta}$.

1 **for** *Task* $K = 1, 2, ...$ **do**
2 $\quad$ **if** $K = 1$ **then**
3 $\quad\quad$ Train the nominal behavior generative model $\phi_1$ with $\mathcal{D}_{\mu_1}$ using Eq. (6)
4 $\quad\quad$ Train the nominal action evaluation model $\boldsymbol{\theta}_1$ with $\mathcal{D}_{\mu_1}$ using Eq. (7)
5 $\quad\quad$ Train the nominal state generative model $\varphi_1$ with $\mathcal{D}_{\mu_1}$ using Eq. (9)
6 $\quad$ **else**
7 $\quad\quad$ Initialize dataset: $\mathcal{D} = \mathcal{D}_{\mu_K}$
8 $\quad\quad$ **for** $k = 1$ **to** $K - 1$ **do**
9 $\quad\quad\quad$ Generate state samples: $\hat{\boldsymbol{s}}_k \sim p_{\varphi_{K-1}}(\boldsymbol{s}|k)$
10 $\quad\quad\quad$ Generate corresponding action samples: $\hat{\boldsymbol{a}}_k \sim \mu_{\phi_{K-1}}(\boldsymbol{a}|\hat{\boldsymbol{s}}_k)$
11 $\quad\quad\quad$ Construct pseudo dataset: $\hat{\mathcal{D}}_k = \sum(\hat{\boldsymbol{s}}_k, \hat{\boldsymbol{a}}_k)$
12 $\quad\quad\quad$ Annotate the Q-function of pseudo state-action pairs as $Q_{\boldsymbol{\theta}_{K-1}}^k(\hat{\boldsymbol{s}}_k, \hat{\boldsymbol{a}}_k)$
13 $\quad\quad\quad$ Interleave pseudo samples with real ones: $\mathcal{D} = \mathcal{D} \cup \hat{\mathcal{D}}_k$
14 $\quad\quad$ **end**
15 $\quad\quad$ Initialization of models: $(\varphi_K, \phi_K, \boldsymbol{\theta}_K) \leftarrow (\varphi_{K-1}, \phi_{K-1}, \boldsymbol{\theta}_{K-1})$
16 $\quad\quad$ Update state generative model $\varphi_K$ with $\mathcal{D}$ using Eq. (10)
17 $\quad\quad$ Update behavior generative model $\phi_K$ with $\mathcal{D}$ using Eq. (13)
18 $\quad\quad$ Update the multi-head critic with $\mathcal{D}$ and $\{Q_{\boldsymbol{\theta}_{K-1}}^k(\hat{\boldsymbol{s}}_k, \hat{\boldsymbol{a}}_k)\}_{k=1}^{K-1}$ using Eq. (15)
19 $\quad$ **end**
20 **end**

---

## APPENDIX B. EXPERIMENTAL SET-UP AND DATA COLLECTION

This section introduces the details of experimental settings for all investigated domains. The problems of interest include: 1) *Swimmer-Dir*, where the swimmer robot needs to move toward a given direction and we randomly sample four target directions in 2D space to form the sequential tasks; 2) *Hopper-Vel*, where the hopper robot needs to run at a goal velocity and we randomly sample four velocities in the range of $[0, 1]$ to form the sequential tasks; 3) *Walker2D-Params*, where a walker robot needs to move forward as fast as possible and we randomly sample four sets of physical parameters for the robot; 4) *HalfCheetah-Vel*, where a planar cheetah needs to run forward at a goal velocity we randomly sample four velocities in the range of $[0, 2]$. For all MuJoCo domains, the time horizon in a learning episode is set as 200.

**Data Collection.** For each evaluation domain, we choose four tasks to execute in a sequence and train a separate policy from scratch to sample the offline dataset for each task. We use soft actor-critic (SAC) (Haarnoja et al., 2018) for the Swimmer-Vel, Hopper-Vel, and Walker2D-Params domains. We use the TD3 algorithm (Fujimoto et al., 2018) for the HalfCheetah-Vel domain as it proves more stable across various HalfCheetah-Vel tasks (Mitchell et al., 2021). The complete replay buffer from the entire lifetime of training is saved for each task and the number of training steps for all tasks is $1M$. Table 3 and Table 4 list the main hyper-parameters for the SAC and TD3 algorithms during offline data collection, respectively.

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

| Parameter | Standard Configuration |
|---|---|
| Optimizer | Adam |
| Value learning rate | $1e-4$ |
| Policy learning rate | $1e-4$ |
| alpha learning rate | $1e-4$ |
| alpha | 0.2 |
| Batch size | 256 |
| Neurons per hidden layer | 256 |
| Number of hidden layers | 1 |
| discount factor | 0.99 |
| target network update rate | 0.01 |

Table 3: Hyperparameters for SAC in the data collection phase.

| Parameter | Standard Configuration |
|---|---|
| Optimizer | Adam |
| Value learning rate | $1e-3$ |
| Policy learning rate | $1e-4$ |
| Batch size | 256 |
| Neurons per hidden layer | 256 |
| Number of hidden layers | 1 |
| discount factor | 0.99 |
| target network update rate | 0.01 |
| frequency of delayed policy updates | 2 |
| range to clip target policy noise | 0.5 |
| policy noise | 0.2 |
| exploration noise | 0.1 |
| frequency of delayed policy updates | 2 |

Table 4: Hyperparameters for TD3 in the data collection phase.

## APPENDIX C. ARCHITECTURE AND HYPERPARAMETERS OF CUGRO

**Network Architecture.** CuGRO includes two conditional scored-based diffusion models that estimate the score function of the behavior action distribution and the score function of the state distribution, respectively, and a multi-head critic model that outputs the Q-values of given state-action pairs. The architecture of the behavior generative model and the state generative model resembles U-Nets, but with spatial convolutions changed to simple dense connections (Janner et al., 2022; Chen et al., 2023). Please refer to Fig. 5 for more details about the network structure. For the multi-head critic model, we use one hidden layer of 256 neurons with SiLU activation functions. We refer to the input and hidden layer as the backbone and the last output layer as the head.

**Hyperparameters.** Table 5 and Table 6 list the main hyperparameters for diffusion models and the multi-head critic used in CuGRO, respectively.

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

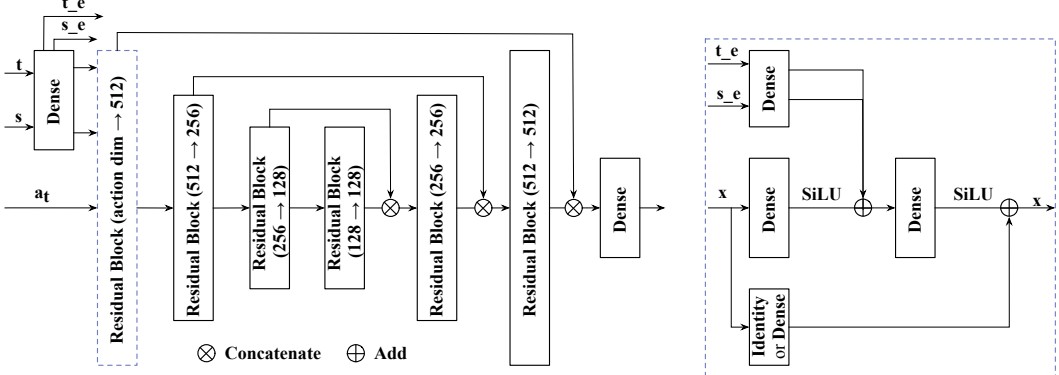

(a) The network architecture of the behavior generative model.

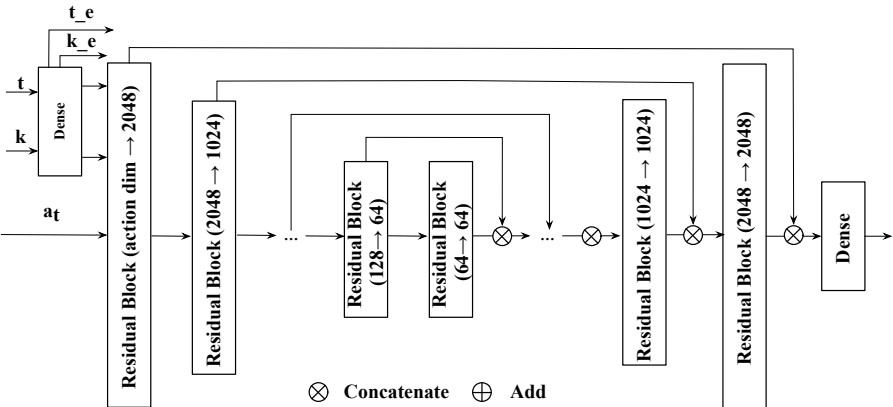

(b) The network architecture of the state generative model.

Figure 5: The network architecture of diffusion models in CuGRO.

| Parameter | Standard Configuration |
|---|---|
| Optimizer | Adam |
| learning rate | $1e-4$ |
| Batch size | 4096 |
| Diffusion steps | 100 |
| epochs | 600 |
| $\beta_{\min}$ | 0.1 |
| $\beta_{\max}$ | 20 |

Table 5: Hyperparameters of Diffusion models.

| Parameter | Standard Configuration |
|---|---|
| Optimizer | Adam |
| learning rate | $1e-4$ |
| Batch size | 4096 |
| epochs | 100 |
| value iteration number | 1 |

Table 6: Hyperparameters of multi-head critic.