# OpenReview forum: "Continual Offline Reinforcement Learning via Diffusion-based Dual Generative Replay"
_ICLR.cc/2024/Conference — Submitted to ICLR 2024_

### Official Review · Reviewer_R6pv · 2023-10-26

**Soundness:** 1 poor
**Presentation:** 2 fair
**Contribution:** 3 good
**Rating:** 3
**Confidence:** 5

**Summary:**

The submission presents a new method for continual offline RL (CORL) that relies on three components: a Q-function to assess the quality of each action, with multiple heads to account for the multiple tasks the agent faces; a diffusion behavior model, to both generate actions to execute in the environment and generate replay data for past tasks to avoid forgetting; and a state diffusion model to generate states to match the distribution observed in the dataset for each past task, again to avoid forgetting. In a strict setting where the agent is not allowed to observe data for any previous task, this enables continual training without forgetting. The authors evaluate their method on four sequences of 4 MuJoCo tasks each, with varying dynamics.

**Strengths:**

######## Strengths ########

- The use of diffusion models both for generating behaviors and replay data is promising
- The overall algorithm carefully leverages the three components above to continually learn new tasks
- The problem of CORL itself is understudied, so it's good to see contributions in this area

**Weaknesses:**

######## Weaknesses ########
- Section 2 (preliminaries) is largely unclear and not stand-alone
- Section 3 (approach) is also not sufficiently clear or precise, and seems to introduce both math and text that are not related to the submission
- The experimental evaluation is not sufficient to assess the benefits of the proposed method

######## Recommendation ########

Unfortunately, I recommend that this manuscript is not accepted in its current form. The main concern I have is the limited experimental evaluation. The authors propose an approach that is a combination of existing ideas (which is fine!), but unfortunately there is not sufficient evidence to support the choice of this combination. Moreover, the draft (especially sections 2 and 3) would need to undergo major revisions to make the text clear and (most importantly) precise.

######## Arguments ########
- Experimental setting -- The place with the most room for improvement in this submission is the empirical evaluation. In particular, I have three main concerns:
    - The evaluation uses very simple MuJoCo tasks to evaluate methods. The authors should evaluate their approach on much more complex tasks, such as those from the Meta-World [1], CausalWorld [2], RLBench [3], or CompoSuite [4,5]. This is particularly true because the authors' motivation stems from the need for diffusion models to represent complex behaviors.
    - The continual setting uses sequences of only 4 tasks. Given that the curves in Figure 2 for CuGRO get progressively worse as more tasks are trained, it would be important to study how this detriment scales with the number of tasks. Can we expect CuGRO to handle a really long stream of tasks? We need empirical evidence to answer that question
    - The evaluation considers no external baselines. There are a number of existing continual learning and offline RL methods (which the authors themselves cite in their manuscript). Yet the evaluation is limited to variations of the authors' proposed method (oracle, noise, and none). Would other continual learning methods be as effective? Would other offline RL methods work well in this setting (especially considering that the tasks are quite simple).
- Clarity of preliminaries
    - The description of advantage-weighted regression misses a key piece: the Q function is w.r.t. \mu, which is the piece that makes Eq. 4 solve the _costrained_ optimization of Eq. 2. This should be explicitly stated and explained, as otherwise the reader might think Q_\theta is the standard Q^\pi --- this is what I thought initially when reading 2.1
    - It's unclear if Eq. 5 corresponds to Eq. 3. I believe that it does, and if so, my understanding is that Q_\theta in Eq. 5 is not the same as Q_\theta in Eq. 4---the former is w.r.t. \pi and the latter is w.r.t. \mu. Please clarify.
    - The description of the diffusion models in Sec 2.2 misses mentioning the existence of a forward diffusion process that adds noise to an (observed) action. This makes it difficult to parse the last sentence before Eq. 6 where the diffusion model is predicting some un-defined noise.
    - The description prior to Eq. 7 is still unclear to me: is Q estimating the value of the actions w.r.t. \mu or \pi? The authors mention that the actions are sampled from \mu, but is the long-term value measured for \pi or \mu? It seems that the middle part of Eq. 7 entails that it's from \pi, but this isn't stated in text.
    - What does re-sampling mean toward the end of Sec 2.1? What was the first "sampling"?

- Clarity/precision of approach
    - Sec 3.1
        - The description of CORL is mostly clear, but I do have a couple of questions:
            - What aspects of the MDP are allowed to change from task to task? If all of them, then which ones do the authors consider in their approach/evaluation?
            - Is the distribution over MDPs P(M) stationary or is it allowed to change over time? Is there any implication of that for the learning process or evaluation setting in the experiments?
    - Sec 3.2
        - While I agree that diffusion models (or generally expressive generative models) are useful for expressing RL policies, the authors seem to conflate two things in their description of why that is the case: 1) some tasks require multimodal behaviors (like in footnote 1) and 2) the overall behavior expressed by the diffusion model should capture a breadth of tasks. The latter is hinted at in both the intro and here, but never actually explained or exemplified.
        - The argument of footnote 2 is somewhat weak. What if two visual classification tasks are "detect if dog is in image" and "detect if cat is in image" and they're both given the same image of a dog? Tasks would require opposite predictions given the same image, just like the RL model would. Plus, the conclusion would be that diffusion models are better because they could capture both the opposite actions, but how is that useful if they can't differentiate when to execute each? The only way to solve a problem like this is to let the model take as input a task indicator (or something to differentiate the tasks), and it's unclear that a Gaussian conditioned on this information would fail.
        - This section is very odd. The first paragraph is all motivation and no technical details. Then second paragraph contains some details about how the diffusion model for state generation works (the equivalent for actions was in the perliminaries). Then the final paragraph is all about doing classifier or classifier-free guidance, but it's unclear what for or why 6 lines of this show up in the middle of a technical section when the authors don't even actually try it. I guess the idea is that the classifier-based/free guidance could ensure that the generated states actually correspond to the conditioning task?
    - Sec 3.3
        - "the desired importance of the new task..." seems to suggest that Eq. 8 should be a weighted sum. But the authors more likely mean that there needs to be some weighting to ensure that the current task is learned sufficiently well while avoiding forgetting. It isn't really about the importance but about being able to optimize properly.
        - It's unclear why there's a test loss (Eq. 11 and 14) for specific models. Isn't performance measured as the obtained reward of the agent on the tasks?
        - It's also unclear what the authors mean by "reconstruct the cumulative state space". Doesn't the task conditioning imply that the agent is learning separate state spaces, one for each task?
        - Are the generated states for behavior replay drawn after updating the state generator? Before? Or are they both updated together? How was this choice made and what are the implications of it? My intuition would be that it's better to first train the behavior model on the fixed state generator and then train the state generator, because updates to the behavior model have no effect on the state generator but the converse is not true.
            - This seems to be clarified in Algorithm 1, but the authors should state it explicitly in text and not rely exclusively on the Appendix to transmit that point.
    - Sec 3.4
        - It's quite unclear after reading this section why the authors use the term behavior cloning, which has a very specific connotation in the context of off-line RL -- replicating the behavior that generates the data.
        - Instead, I think this approach is better described as a form of _functional regularization_, which is a method broadly studied in supervised continual learning research.

[1] Yu et al., "Meta-World: A benchmark and evaluation for multi-task and meta reinforcement learning." CoRL, 2020

[2] Ahmed et al., "CausalWorld: A Robotic Manipulation Benchmark for Causal Structure and Transfer Learning." ICLR, 2021

[3] James et al., "Rlbench: The robot learning benchmark & learning environment." RAL, 2020

[4] Mendez et al., "CompoSuite: A Compositional reinforcement learning benchmark." CoLLAs, 2022

[5] Hussing et al., "Robotic Manipulation Datasets for Offline Compositional Reinforcement Learning." arXiv, 2023

**Questions:**

######## Additional feedback ########

The following points are provided as feedback to hopefully help better shape the submitted manuscript, but did not impact my recommendation in a major way.

Abstract
- The abstract is very clear. It lays out very well how the approach works and the results they obtain

Intro
- It's unclear what "new tasks emerge overwhelmingly" means or why van de Ven et al. is cited to support that claim.
- The motivation for why CORL is special seems to be all about RL in general, and not specifically about offline RL. This seems to undermine the need to develop specialized approaches.
- There's always the question of whether the size of the model might surpass the size of replay buffers. In visual settings that tends to happen. It's unclear if it does here
- The idea of using generative models to express state and action distributions (which isn't novel) is good, especially in the offline setting where we can't make assumptions about the form of the distribution
- What does "behavior cloning matter" mean in the context of a critic, which is not a behavior model?

Sec 2
- I appreciate the notation clarification right before Sec 3! The authors could consider moving it to the beginning of Sec. 2 so the reader knows this ahead of time.
- I thought we were missing a description of CORL, but that's in Sec. 3. Is the formalization of CORL a contribution of this work? If not, maybe it's worth also including it in Sec 2.

Sec 3.1
- It does seem like the problem setting should be moved to Sec 2, and then the overview be placed in Sec 3 before introducing Sec 3.2 (which would be 3.1) below
- Fig. 1 is useful. Why are there two (s,a,r,s') boxes in b and c? It seems like they are the same tuple, just that the first down arrow takes s and the second down arrow takes a.

Sec 4
- "the generator is the only constraint on task performance" -- how is this an improvement over other continual learning methods?
- "When the generative model is optimal, training the networks with generative replay is equivalent to joint training on the entire dataset."
    - Sure, an optimal replay method would achieve that... but can we actually train an optimal generator over a long sequence of tasks? Also, it is only equivalent to joint training if we actually do full joint training (from scratch), but not if we start from the previously trained models. Starting from previously trained models might be better or worse, but certainly not equivalent.
- What is the (final) performance of SAC/TD3 on the collected datasets?
- What is the "oracle" for the behavior model? And what is the "noisy" replay for the behavior model?
- More than baselines, these seem to be ablations of CuGRO. While oracle is roughly a performance upper bound, it's unclear how other existing continual learning algorithms would perform compared to CuGRO/oracle. It's also unclear if non-diffusion approaches (given "oracle" or some other forgetting avoidance method) would work well.
- I like the analysis of why Hopper-Vel fails
- In Figure 3 we don't get to see past task performance. Or is this average performance including current and past tasks? Does Table 1 measure the final performance of all past tasks after training on the final task, or upon finishing training on each individual task?
- The hyperparamter sensitivity analysis is good and useful.

Typos/style/grammar
- Footnotes in 3.2 should go before periods, not after the period and a space (e.g., "...keep emerging\footnote{text}.")
- Sec 3.2, paragraph 1 -- do datasets really "emerge"? Maybe they are "constructed" instead
- Sec 3.2, paragraph 2 -- scored-based --> score-based
- Sec 3.3, first line -- technically, the models are \mu_\phi and \epsilon_\psi, and \phi eand \psi are the parameters
- I find the use of "replayed" samples (throughout the text) a bit odd, since they aren't real samples. I'd suggest using "generated" samples instead to consistently clarify that these are not real.

---

> ### Author Response · Authors · 2023-11-18
> **Response (Part 1/3)**
>
> **Q1. Experiments: i) Comparison to external baselines of existing continual learning and offline RL methods; ii) Evaluation on more testbeds like Meta-World, CausualWorld, RLBench, or CompoSuite;  iii) CuGRO's scalability with the number of tasks in continual learning.**
>
> A1: i) As we are the first to leverage expressive diffusion models to tackle the understudied CORL challenge, our primary concern is to validate whether our dual generative replay mechanism can realize high-fidelity replay of the sample space.
> Hence, following the classical DGR method [1], we compare the performance of the RL model trained with variants of replay methods: Oracle, None, and Noise.
> For Oracle, the generated data is replaced with real data of past tasks to demonstrate that our diffusion-based generator can mimic the data distribution with high-fidelity.
> For Noise, the state generator is replaced with Gaussian noise to demonstrate the effectiveness of our diffusion-based state generator.
> For None, the state generator is removed and catastrophic forgetting is incurred, which demonstrates the necessity of our generative replay mechanism.
> We did not compare our method to several existing continual RL methods due to the difference in the intrinsic characteristics and applicable scenarios.
> For example, [2] belongs to the parameter isolation kind, [3, 4] needs to replay real samples of past tasks, and most of the existing methods [2, 3, 5] tackle the online RL problems other than the offline setting.
>
> ii) These testbeds are usually used to test traditional online RL algorithms and has not been applied to offline scenarios yet.
> Compared with online continual RL, offline continual RL is less studied, and there is no standard dataset for it.
> Generally, the MuJoCo environment, as seen in the D4RL dataset [6], is a popular testbed for offline RL.
> The MuJoCo environment is also commonly used in offline meta-RL for testing [7, 8].
> Hence, we evaluate CORL algorithms on MuJoCo environments with offline datasets in this paper.
>
> iii) We follow the generative replay paradigm, e.g., DGR [1] and DDGR[9], where the scalability primarily depends on the modeling fidelity of the generator.
> This rationale underpins the choice of diffusion models for our method, mirroring the research motivation of DDGR.
> In DDGR, diffusion models demonstrate better fidelity compared to GANs, and exhibit promising superiorities for continual learning based on generated replay.
> Our method is the first step that leverages expressive diffusion models to tackle the CORL challenge, and we believe that this approach will see further developments in the future.
>
> In summary, it is important to evaluate CORL methods on more testbeds, or to conduct empirical comparisons to existing methods for demonstrating their respective pros and cons, or to modify these online continual methods to adapt to offline settings.
> We leave these for future work.
>
>
> [1] Hanul Shin, et al., Continual learning with deep generative replay, NeurIPS 2017.
>
> [2] Gaya et al., Building a subspace of policies for scalable continual learning, ICLR 2023.
>
> [3] Maciej Wolczyk, et al., Disentangling transfer in continual reinforcement learning, NeurIPS 2022.
>
> [4] Sibo Gai, et al., OER: Offline experience replay for continual offline reinforcement learning, ECAI 2023.
>
> [5] David Rolnick, et al., Experience replay for continual learning, NeurIPS 2019.
>
> [6] Justin Fu, et al., D4RL: Datasets for Deep Data-Driven Reinforcement Learning, arXiv:2004.07219, 2020.
>
> [7] Eric Mitchell, et al., Offline meta-reinforcement learning with advantage weighting, ICML 2021.
>
> [8] Haoqi Yuan, et al., Robust task representations for offline meta-reinforcement learning via contrastive learning, ICML 2022.
>
> [9] Rui Gao, et al., DDGR: Continual learning with deep diffusion-based generative replay, ICML 2023.

---

> ### Author Response · Authors · 2023-11-18
> **Response (Part 2/3)**
>
> **Q2. Clarity of approach: i) Change of the MDP distribution from task to task; ii) Claim of multimodal behaviors within one task and across tasks; iii) The weighting of losses across tasks; iv) About test loss in Eqs. (11) and (14); v) The order of training model components; vi) About behavior cloning.**
>
> A2: i) One task corresponds to an MDP, and tasks are generated from a specific MDP distribution in continual learning. We have explained the details of sequential tasks in Sec. 3.1 and Appendix B.
>
> ii) The behavior policy of a single task may be a multimodal distribution.
> When considering multiple tasks, the distribution of all behavior policies might become even more multimodal.
> Hence, we use a diffusion model to fit the behavior policies of all tasks.
> In classification tasks, opposite predictions are resolved using a task indicator.
> Similarly, in RL, the diffusion model mimics behavioral strategies, while the decision-making is guided by the Q-function.
>
> iii) In Eqs. (10) and (13), we present the general form of training loss and set $\beta$ to facilitate training.
> During the training process, the value of $\beta$ is 0.5.
>
> iv) The goal of  Eq. (8) is to learn a continual policy that maximizes the expected return over all encountered tasks, which aligns with the training objectives of Eqs. (10) and (13).
> The test loss in Eqs. (11) and (14) evaluates the training effectiveness of the diffusion model, analogous to the validation loss in supervised learning.
> The setting of the test loss is consistent with DGR [1].
>
> v) We decouple the policy into a behavior generative model and an action evaluation model, and use the state generative model along with the behavior generative model to achieve generative replay.
> During the continual learning process, we use a continually updated state generation model to cover the state space of all tasks, enabling it to reconstruct the cumulative state space of all tasks based on task encoding.
> The order of training model components is actually shown in Fig. 1.
>
> vi) In our method, action sampling is guided by Q-values, analogous to classical value function methods where the policy/behavior is derived from value functions.
> Hence, cloning the Q-function implicitly clones the behaviors, thus we call the process as behavior cloning.
>
>
> **Q3: Clarity of preliminaries: i) The description of advantage-weighted regression; ii) The relationship of Eqs. (3)-(5); iii) The description of the diffusion models in Sec 2.2 misses mentioning the existence of a forward diffusion process that adds noise to an (observed) action; iv) The description prior to Eq. (7); v) What does re-sampling mean toward the end of Sec 2.1?**
>
> A1: i) Following your suggestions, we have further explained $Q_\theta$ and $Q^\pi$ and their relationship.
>
> ii) Eq. (4) is the general modeling form of offline RL, Eq. (5) is the decoupling form proposed by SfBC [10], and $Q_\theta$ is the same in Eqs. (4) and (5).
>
> iii) In Section 2.2, we describe the forward diffusion process of the diffusion model. Diffusion models are generative models that begin by defining a forward process to gradually add noise to an unknown data distribution, and then learn to reverse this process.
>
> iv) Eq. (7) is based on the fundamental concepts of offline RL.
> $R_i^{(0)}$ is the vanilla return of trajectories.
> The first part of Eq. (7) offers an implicit planning scheme within dataset trajectories that mainly helps to avoid bootstrapping over unseen actions and accelerate convergence.
> The middle part of Eq. (7) enables the generalization of actions in similar states across different trajectories.
> This is our basic approach, and you can refer to SfBC for details [10].
>
> v) The 'first' sampling means sampling from the diffusion-based behavior generative model.
> Then, we evaluate the sampled actions with the trained critic network.
> The learned policy is a combination of the behavior generative model and the critic network.
> Finally, during execution, actions are 're-sampled' from the learned policy.
> More details can be found in [10].
>
>
> [10] Huayu Chen, et al., Offline reinforcement learning via high-fidelity generative behavior modeling, ICLR 2023.

---

> ### Author Response · Authors · 2023-11-18
> **Response (Part 3/3)**
>
> **Q4: Introduction: i) What new tasks emerge overwhelmingly' means; ii) The motivation for why CORL is special seems to be all about RL in general, and not specifically about offline RL; iii) There's always the question of whether the size of the model might surpass the size of replay buffers.**
>
> A1: i) It means that in a continual learning setting, new tasks keep emerging in a sequential manner.
>
> ii) Our method does not operate in an online mode and is solely suitable for offline scenarios.
> Under the offline RL paradigm, we model behavior policies using a diffusion model.
>
> iii) The size of the model will not exceed the size of the replay buffer.
> Taking the *HalfCheetah-Vel* environment as an example, we only require an additional state generative model to realize generative replay.
> The state generator has about 24M parameters, and each new task has about 1M samples where each sample ($s,a$) has a dimension of 26.
> As the number of tasks increases, the size of the replay buffer will continue to expand, but the model size will remain constant.
> Moreover, we assume that the data from previous tasks is not available, a challenge often encountered in real-world scenarios, making generative replay necessary.
>
>
> **Q5: Sec. 2 and Sec 3.1: i) The authors could consider moving the notation clarification to the beginning of Sec. 2; ii) Move the description of CORL in Sec 2; iii) Why are there two ($s,a,r,s'$) boxes in Figs. 1(b) and 1(c)?**
>
> A5: i) Following your suggestion, we have moved the notation clarification to the beginning of Sec. 2.
>
> ii) Additionally, we have described CORL in Section 3.1 and have now relocated it to Sec. 2.
>
> iii) In Fig. 1, in order to clearly show that real samples and generated samples are used together to train the behavior evaluation model, we use two ($s, a, r, s'$)  boxes to correspond to the state generator box and the behavior generator box, respectively.
>
>
>
> **Q6: Sec. 4: i) "The generator is the only constraint on task performance" -- how is this an improvement over other continual learning methods? ii) Starting from previously trained models might be better or worse, but certainly not equivalent to joint training on the entire dataset; iii) What is the (final) performance of SAC/TD3 on the collected datasets? iv) Explanations of baselines**
>
> A6: i) Continual learning methods can mainly be divided into three categories: regularization-based, parameter isolation, and rehearsal methods.
> We focus on the rehearsal method, using experience replay, and propose using a generative model to generate replay samples rather than storing samples of old tasks.
> Therefore, the generator is the only constraint on task performance in our method.
> Compared with methods that need to store samples of old tasks, our method only requires storing the generative model, which greatly saves memory space.
>
> ii) In the continual learning process, we train the networks using generated samples of old tasks and real samples of new tasks together.
> If the generative model based on the diffusion model is good enough, it can generate very accurate samples, equivalent to joint training based on real data from all current tasks.
>
> iii) The final performance of SAC/TD3 on the collected dataset is shown in the table below.
>
> | Swimmer-Dir     | 1048.38 |
> | Hopper-Vel      | 185.72  |
> | Walker2D-Params | 164.88  |
> | HalfCheetah-Vel | -24.75  |
>
>
> iv) We have explained our baseline in Section 4.
> Oracle uses real samples of all previous tasks to sequentially train the dual generators, and to train the multi-head critic with behavior cloning.
> *Noise* considers the opposite case when generated samples do not resemble the real distribution at all.
> We omit the state generative model and use random noises as replayed state samples.
> The behavior generator and multi-head critic are sequentially trained using these randomly generated states.
> Oracle stores real samples of all old tasks, which is the best case in the replay method, and the performance of our method is very close to Oracle.
> Additionally, from the experimental results of 'None' and 'Noise,' we can see that the performance decreases when the state generative model is removed or there is no replay.
> Figure 3 shows the average performance including current and past tasks.
> Table 1 shows the final performance on all past tasks after training on the final task.
>
>
> **Q7:  Typos/style/grammar issues.**
>
> A7: We sincerely thank the reviewer for their careful reading and have revised the paper according to your suggestions.

---

> > ### Comment · Reviewer_R6pv · 2023-11-18
> > **Thank you for your response**
> >
> > I thank the authors for their response. I have read it in detail and unfortunately did not find sufficient evidence to update my score. Critically, the authors defend the lack of need to expand their experimental setting, which I believe to be critical for making this into a fully developed contribution.

---

### Official Review · Reviewer_t1vB · 2023-10-30

**Soundness:** 3 good
**Presentation:** 3 good
**Contribution:** 2 fair
**Rating:** 5
**Confidence:** 3

**Summary:**

The authors in the paper propose a dual generative reply framework to address the challenges in continual RL, where the practical algorithms are required to adapt to new environments and simultaneously leverage the previous knowledge. In particular, they use two diffusion models to generate both the state and behavior generative reply on the offline data and update in a sequential way. Using the behavior cloning technique, the multi-head critic is updated effectively, and the resulting algorithm CuGRO suggests competitive performance in the considered Mujoco benchmark.

**Strengths:**

* The paper is well-organized and easy to follow.
* The proposed method is technically sound, including the incorporation of diffusion models to mimic the generative buffer reply.
* The empirical performance seems significant compared with considered baselines.

**Weaknesses:**

* Incorporating diffusion models in offline RL for generative reply is straightforward, and the computation cost should be rigorously discussed.
* The loss function in Eq.10, 11, 13, 14, and 15 is easy to figure out, but it is hard to guarantee the convergence. For instance, in Eq.10, it is not clear whether training a diffusion model via using the data from the last diffusion model has any convergence guarantee. This issue could be severe especially when we only have access to offline data with a limited size or large state and action space.
* Missing other continual RL baselines or benchmarks. In continual RL, the benchmark ‘continual world’ is commonly used to evaluate the continual control algorithms, but the proposal algorithm is only evaluated in small-scaled benchmarks in Mujuco games. It is not clear whether this approach is effective in the commonly-used continual RL environments. Also, some typical continual RL baselines are missing, such as [1].

[1] Gaya et al. Building a subspace of policies for scalable continual learning (ICLR 2023)

**Questions:**

Please refer to Weakness.

Overall, as far as I can tell, the proposed method is technically sound and achieves competitive performance. However, the computation cost would be large, and it also lacks discussion about the convergence guarantee and training details. Some typical benchmarks and baselines should be considered as well.

---

> ### Author Response · Authors · 2023-11-17
> **Response (Part 1/2)**
>
> **Q1: Incorporating diffusion models in offline RL for generative reply is straightforward, and the computation cost should be rigorously discussed.**
>
> A1: In the continual learning setting, the category of generative replay methods [1, 2] needs to maintain two separate models: the solver (the backbone RL algorithm) and the generator (which mimics the data distribution of past tasks).
> That is, they eliminate the dependency on accessing data of past tasks at the cost of maintaining an additional generative model, e.g., GANs [1] or diffusion models [2].
> Taking the HalfCheetah task as an example, the state generator has about 24M parameters, and each new task has about 1M samples where each transition sample ($s,a,r,s'$) has a dimension of 26.
> Training the state generator with 1M samples roughly costs 2.5 hours on an NVIDIA RTX 3090 GPU, which could be further reduced to 50 minutes via distributed training on 4 GPUs.
>
> In literature, current diffusion-based RL methods [3-8] could face the challenge of increased computational costs due to the training and sampling of diffusion models.
> For example, the diffusion-planning methods [4, 8] also need to train two separate diffusion models, one to generate the trajectory data and another to predict the cumulative rewards of trajectory samples.
> We believe that the fast development of diffusion models in the generative modeling community can help tackle the above challenge for RL problems, such as sampling acceleration methods [9, 10].
> Our method is the first step that leverages expressive diffusion models to tackle the CORL challenge, and we will continue to tackle the limitations in future work as stated in the original paper (bottom, page 9): "Though, our method requires two diffusion models to synthesize replayed samples, which could be further improved by sampling acceleration methods or developing one diffusion model for unifying the state and behavior modeling. We leave these directions as future work."
>
>
> **Q2: The convergence guarantee, in Eqs. (10)-(11) and (13)-(15), of training a diffusion model via using the data from the last diffusion model, especially when we only have access to offline data with a limited size or large state and action space.**
>
> A2: Under the generative replay paradigm [1, 2], we train the diffusion models using a mixed distribution of real data from the new task and pseudo data generated from previous generators.
> The convergence regarding Eqs. (10), (11), (13), and (14) align with the diffusion model.
> In Eq. (15), the first term follows a planning-based Bellman operator used in offline RL [3], and the second term corresponds to behavior cloning.
> Hence, the convergence regarding Eq. (15) is consistent with the specific Bellman operator [3] and the supervised stochastic gradient descent (SGD).
> The convergence of diffusion models is independent of the training data.
> Continually training the diffusion model with data generated by the previous iteration of the model only affects the quality of the training data and does not influence the model's convergence.
> Hence, the convergence remains consistent with the vanilla diffusion models.
> Additionally, the convergence of the model is independent of the dataset size.
>
> [1] Hanul Shin, et al., Continual learning with deep generative replay, NeurIPS 2017.
>
> [2] Rui Gao, et al., DDGR: Continual learning with deep diffusion-based generative replay, ICML 2023.
>
> [3] Huayu Chen, et al., Offline reinforcement learning via high-fidelity generative behavior modeling, ICLR 2023.
>
> [4] Michael Janner, et al., Planning with diffusion for flexible behavior synthesis, ICML 2022.
>
> [5] Zhendong Wang, et al., Diffusion Policies as an Expressive Policy Class for Offline Reinforcement Learning, ICLR 2023.
>
> [6] Bingyi Kang, et al., Efficient Diffusion Policies for Offline Reinforcement Learning, NeurIPS 2023.
>
> [7] Anurag Ajay, et al., Is Conditional Generative Modeling all you need for Decision Making? ICLR 2023.
>
> [8] Fei Ni, et al., MetaDiffuser: Diffusion Model as Conditional Planner for Offline Meta-RL, ICML 2023.
>
> [9] Cheng Lu, et al., DPM-Solver: A Fast ODE Solver for Diffusion Probabilistic Model Sampling in Around 10 Steps, NeurIPS 2022.
>
> [10] Qinsheng Zhang, et al., gDDIM: Generalized denoising diffusion implicit models, ICLR 2023.

---

> ### Author Response · Authors · 2023-11-17
> **Response (Part 2/2)**
>
> **Q3: Missing continual RL benchmarks (e.g., continual world) and baselines (e.g., [11]).**
>
> A3: As we are the first to leverage expressive diffusion models to tackle the understudied CORL challenge, our primary concern is to validate whether our dual generative replay mechanism can realize high-fidelity replay of the sample space.
> Hence, following the classical DGR method [1], we compare the performance of the RL model trained with variants of replay methods: Oracle, None, and Noise.
> For Oracle, the generated data is replaced with real data of past tasks to demonstrate that our diffusion-based generator can mimic the data distribution with high-fidelity.
> For Noise, the state generator is replaced with Gaussian noise to demonstrate the effectiveness of our diffusion-based state generator.
> For None, the state generator is removed and catastrophic forgetting is incurred, which demonstrates the necessity of our generative replay mechanism.
> We did not compare our method to several existing continual RL methods due to the difference in the intrinsic characteristics and applicable scenarios.
> For example, [11] belongs to the model expansion kind, [12, 13] needs to replay real samples of past tasks, and most of the existing methods [11, 12, 14] tackle the online RL problems other than the offline setting.
>
> 'Continual world' is currently used to test traditional online RL algorithms and has not been applied to offline scenarios yet.
> Compared with online continual RL, offline continual RL is less studied, and there is no standard dataset for it.
> Generally, the MuJoCo environment, as seen in the D4RL dataset [16], is a popular testbed for offline RL.
> The MuJoCo environment is also commonly used in offline meta-RL for testing [8, 16, 17].
> Hence, we evaluate CORL algorithms on MuJoCo environments with offline datasets in this paper.
>
> On the other hand, it is important to evaluate CORL methods on more testbeds like the continual world in the future.
> It is also significant to conduct empirical comparisons to existing methods for demonstrating their respective pros and cons, or to modify these online continual methods to adapt to offline settings.
> We leave these for future work.
>
>
> [1] Hanul Shin, et al., Continual learning with deep generative replay, NeurIPS 2017.
>
> [8] Fei Ni, et al., MetaDiffuser: Diffusion Model as Conditional Planner for Offline Meta-RL, ICML 2023.
>
> [11] Gaya et al., Building a subspace of policies for scalable continual learning, ICLR 2023.
>
> [12] Maciej Wolczyk, et al., Disentangling transfer in continual reinforcement learning, NeurIPS 2022.
>
> [13] Sibo Gai, et al., OER: Offline experience replay for continual offline reinforcement
> learning, ECAI 2023.
>
> [14] David Rolnick, et al., Experience replay for continual learning, NeurIPS 2019.
>
> [15] Justin Fu, et al., D4RL: Datasets for Deep Data-Driven Reinforcement Learning, arXiv:2004.07219, 2020.
>
> [16] Eric Mitchell, et al., Offline meta-reinforcement learning with advantage weighting, ICML 2021.
>
> [17] Haoqi Yuan, et al., Robust task representations for offline meta-reinforcement learning via contrastive learning, ICML 2022.

---

### Official Review · Reviewer_RUFG · 2023-10-30

**Soundness:** 2 fair
**Presentation:** 3 good
**Contribution:** 2 fair
**Rating:** 5
**Confidence:** 4

**Summary:**

The paper addresses the continual offline reinforcement learning (CORL) problem, focusing on the challenge of catastrophic forgetting as models encounter new tasks. To combat this, the study introduces CuGRO, a method that decouples the learning policy into a generative behavior model and an action evaluation model, ensuring diverse behaviors are captured. A state generative model is also employed to mimic past task distributions. By leveraging diffusion probabilistic models, CuGRO achieves high-fidelity sample reproduction. Empirical tests reveal CuGRO's superiority in reducing forgetting and enhancing forward transfer, closely matching results using original data.

**Strengths:**

1. Innovative idea: The paper introduces CuGRO, a novel framework of CORL, which is one of the first to leverage expressive diffusion models for this challenge.
2. Practical memory solutions: Instead of relying on large buffers to store real samples from prior tasks, CuGRO synthesizes high-quality pseudo-samples, addressing the challenges of memory constraint and potential privacy issues.
3. Empirical validation: The paper provides empirical evidence from various tasks that demonstrates CuGRO's effectiveness in mitigating forgetting.

**Weaknesses:**

1. Motivation. There is no sufficient support for the necessity of using diffusion models to learn the behaviors from prior tasks. Though the general knowledge is that diffusion models can work better in terms of generation and generalization, there is no explicit reason against the utilization of other modes such as behavior cloning, GAN, VAE, etc.
2. Efficiency trade-off: Though storing models for prior tasks might work well, training diffusion models for each different task might not be sample-efficient nor computation-efficient. Therefore, the authors might want to provide more information on the feasibility of this approach and the computation resource usage for implementing the experiments.
3. Experiment: The experiment did not show how diffusion models contribute to the continual learning of the model. The authors might want to show the performance of the diffusion models to demonstrate that diffusion models are contributing to the performance. In addition, it is unclear in the paper regarding the training data of the diffusion models. Will noisy data degrade the performance of the diffusion models and the CuGRO model as a whole?

**Questions:**

1. I was wondering if applying other types of generative models to replace the diffusion model will yield similar performance.
2. I am curious to know the computation resources used and how long to train the model.
3. I was wondering about the scalability of the model. If having more tasks degrade the model's performance?

---

> ### Author Response · Authors · 2023-11-17
> **Response (Part 1/2)**
>
> **Q1: On the necessity of using diffusion models to learn behaviors from prior tasks, and the reason against the utilization of other models such as behavior cloning, GAN, VAE, etc. If applying other types of generative models to replace the diffusion model will yield similar performance?**
>
> A1: As repeatedly stated in our original paper, we have discussed the advantages of the diffusion model and the motivation for adopting it to tackle the CORL challenge.
> Due to the strong distributional expressivity and the ability to fit multimodal distributions, diffusion models have been explosively developed in the RL community recently [1-5], such as using diffusion models for behavior policy modeling [1], for target policy modeling [2], and for model-based trajectory modeling [2].
> In the first paragraph of Sec. 3.2 (bottom, page 4), we elaborate on the necessity of using the diffusion model to learn behavior from prior tasks as: "This perspective rooted in generative modeling presents three promising advantages for CORL. First, existing policy models are usually unimodal Gaussians with limited distributional expressivity, while collected behaviors in CORL become progressively diverse as novel datasets keep emerging. Learning a generative behavior model is considerably simpler since sampling from the behavior policy can naturally encompass a diverse range of observed behaviors, and allows the policy to inherit the distributional expressivity of diffusion models. Second, RL models are more prone to deficient generalization across diverse tasks. Learning a unified behavior model can naturally absorb novel behavior patterns, continually promoting knowledge transfer and generalization for offline RL. Third, generative behavior modeling can harness extensive offline datasets from a wide range of tasks with pretraining, serving as a foundation model to facilitate finetuning for any downstream tasks. This aligns with the paradigm of large language models, and we reserve this promising avenue for future research."
>
> On the other hand, other models such as behavior cloning, GANs, and VAEs can also be considered as alternatives to the diffusion model.
> In this paper, we choose the powerful diffusion model due to its promising properties discussed above.
> As the state-of-the-art architecture in the generative modeling community, the diffusion model has achieved great success in high-fidelity data generation, such as its superiority over GANs [6] and stable diffusion [7].
>
> **Q2:  Experiment: i) Show how diffusion models contribute to continual learning performance; ii) Clarify the training data of the diffusion models; iii) Will noisy data degrade the performance of the diffusion models and the CuGRO model as a whole?**
>
> A2: i) As stated in our experiments, our primary concern is to validate whether our diffusion-based generative replay mechanism can realize high-fidelity replay of the sample space.
> Hence, we set three baselines as Oracle, None, and Noise.
> For Oracle, the generated data is replaced with real data of past tasks to demonstrate that our diffusion-based generator can mimic the data distribution with high-fidelity.
> For Noise, the state generator is replaced with Gaussian noise to demonstrate the effectiveness of our diffusion-based state generator.
> For None, the state generator is removed and catastrophic forgetting is incurred, which demonstrates the necessity of our generative replay mechanism.
> In summary, these observations verify the contributions of our diffusion models to the continual learning performance.
>
> ii) The collection details of offline datasets are explained in Appendix B.
> As stated in Sec. 3.3, the diffusion models are trained using a mixed data distribution of real samples and replayed ones from the previous generator.
>
> iii) We only use the noisy data in the Noise baseline, where the state generative model is replaced with Gaussian noise.
> In CuGRO, we train the diffusion models with a mixed distribution of real data from the new task and pseudo data generated from previous generators.
>
> [1] Huayu Chen, et al., Offline reinforcement learning via high-fidelity generative behavior modeling, ICLR 2023.
>
> [2] Zhendong Wang, et al., Diffusion Policies as an Expressive Policy Class for Offline Reinforcement Learning, ICLR 2023.
>
> [3] Michael Janner, et al., Planning with diffusion for flexible behavior synthesis, ICML 2022.
>
> [4] Bingyi Kang, et al., Efficient Diffusion Policies for Offline Reinforcement Learning, NeurIPS 2023.
>
> [5] Anurag Ajay, et al., Is Conditional Generative Modeling all you need for Decision Making? ICLR 2023.
>
> [6] Prafulla Dhariwal et al., Diffusion models beat GANs on image synthesis, NeurIPS 2021.
>
> [7] Robin Rombach et al., High-Resolution Image Synthesis with Latent Diffusion Models, CVPR 2022.

---

> ### Author Response · Authors · 2023-11-17
> **Response (Part 2/2)**
>
> **Q3: Efficiency trade-off: Training diffusion models for each different task might not be sample-efficient nor computation-efficient. Therefore, the authors might want to provide more information on the feasibility of this approach and the computation resource usage for implementing the experiments (e.g., how long to train the model).**
>
> A3: In the continual learning setting, the category of generative replay methods [8, 9] needs to maintain two separate models: the solver (the backbone RL algorithm) and the generator (which mimics the data distribution of past tasks).
> That is, they eliminate the dependency on accessing data of past tasks at the cost of maintaining an additional generative model, e.g., GANs [8] or diffusion models [9].
>
> **For sample efficiency**, our method operates in an offline mode, and the size of the offline dataset is fixed.
> Hence, the sample efficiency is not a primary consideration.
>
> **For computation efficiency**, taking the HalfCheetah task as an example, the state generator has about 24M parameters, and each new task has about 1M samples where each transition sample ($s,a,r,s'$) has the dimension of 26.
> As the number of tasks increases, the number of encountered samples will continue to expand, while the model size remains constant.
> Training the state generator with 1M samples roughly costs 2.5 hours on an NVIDIA RTX 3090 GPU, which could be further reduced to 50 minutes via distributed training on 4 GPUs.
> The advantage of generative replay becomes more remarkable for scenarios where massive datasets are needed to train a task or the continual learning setting contains a large number of sequential tasks.
> Additionally, the generative replay method can produce pseudo data as much as needed, which can also improve the robustness and flexibility of the model.
>
> In literature, current diffusion-based RL methods [1-5] could face the challenge of increased computational costs due to the training and sampling of diffusion models.
> For example, the diffusion-planning methods [3] also need to train two separate diffusion models, one to generate the trajectory data and another to predict the cumulative rewards of trajectory samples.
> We believe that the fast development of diffusion models in the generative modeling community can help tackle the above challenge for RL problems, such as sampling acceleration methods [10, 11].
> Our method is the first step that leverages expressive diffusion models to tackle the CORL challenge, and we will continue to tackle the limitations as stated in the original paper (bottom, page 9): "Though, our method requires two diffusion models to synthesize replayed samples, which could be further improved by sampling acceleration methods or developing one diffusion model for unifying the state and behavior modeling. We leave these directions as future work."
>
>
> **Q4:  I was wondering about the scalability of the model. If having more tasks degrade the model's performance?**
>
> A4: We follow the generative replay paradigm, e.g., DGR [8] and DDGR[9], where the scalability primarily depends on the modeling fidelity of the generator.
> This rationale underpins the choice of diffusion models for our method, mirroring the research motivation of DDGR.
> In DDGR, diffusion models demonstrate better fidelity compared to GANs, and exhibit promising superiorities for continual learning based on generated replay.
> Our method is the first step that leverages expressive diffusion models to tackle the CORL challenge, and we believe that this approach will see further developments in the future.
>
> [1] Huayu Chen, et al., Offline reinforcement learning via high-fidelity generative behavior modeling, ICLR 2023.
>
> [2] Zhendong Wang, et al., Diffusion Policies as an Expressive Policy Class for Offline Reinforcement Learning, ICLR 2023.
>
> [3] Michael Janner, et al., Planning with diffusion for flexible behavior synthesis, ICML 2022.
>
> [4] Bingyi Kang, et al., Efficient Diffusion Policies for Offline Reinforcement Learning, NeurIPS 2023.
>
> [5] Anurag Ajay, et al., Is Conditional Generative Modeling all you need for Decision Making? ICLR 2023.
>
> [8] Hanul Shin, et al., Continual learning with deep generative replay, NeurIPS 2017.
>
> [9] Rui Gao, et al., DDGR: Continual learning with deep diffusion-based generative replay, ICML 2023.
>
> [10] Cheng Lu, et al., DPM-Solver: A Fast ODE Solver for Diffusion Probabilistic Model Sampling in Around 10 Steps, NeurIPS 2022.
>
> [11] Qinsheng Zhang, et al., gDDIM: Generalized denoising diffusion implicit models, ICLR 2023.

---

### Official Review · Reviewer_msqf · 2023-11-02

**Soundness:** 2 fair
**Presentation:** 3 good
**Contribution:** 3 good
**Rating:** 6
**Confidence:** 2

**Summary:**

Motivated by the limitations of unimodal Gaussian policy models and the memory constraints of storing data from previous tasks, the authors propose a novel dual generator system to facilitate continual learning in reinforcement learning. This system features a behavior generative model that diffuses over actions given states, and a state generative model that diffuses over states from past tasks without needing to store all previous data. When encountering a new task, this dual approach leverages the state generator to generate synthetic state samples reflective of all former tasks, and the behavior generator produces corresponding actions, forming pseudo state-action pairs.
Than, a multi-head critic network, with separate heads dedicated to individual tasks, is trained on real samples from new datasets and annoate the pseudo pairs to create synthetic samples for behavior cloning.

**Strengths:**

Pros:
1. First attempt to incorporate diffusion model for continual offline RL. Novel idea to utilizing diffusion-model’s expressiveness to generate high-fidelity replay of the previous tasks to prevent the need of storing all previous tasks samples.
2. Shows effectiveness in generating new samples to represent prior tasks when comparing CuGRO with the Oracle in Table 1 and figure 2.
3. Achieves strong experimental results across the 4 simulated environments. The proposed CuGRO algorithm closely matches the Oracle, outperforming baselines.
4. Ablation studies conducted to analyze the hyper parameters lambda which controls how much emphasis to put on the previously replayed dataset.

**Weaknesses:**

Cons:
1. Requires training two separate diffusion models, which can be computationally expensive for sampling at test time since parallel sampling is not possible. Exploring concatenating {s,a} and diffuse with one model could improve efficiency.
2. This methods alleviates the memory capacity concern by condensing the previous task’s knowledge into two diffusion models. How does the continual training cost of updating diffusion models for each new task and sampling from them trade off with the memory savings of condensing previous tasks?
3. Limited baselines: Is comparison only made within the diffusion-based model generator pipeline? Why there is no comparison to previous continual RL algorithms provided?

**Questions:**

please see weaknesses

---

> ### Author Response · Authors · 2023-11-17
> **Response (Part 1/2)**
>
> **Q1: Training two separate diffusion models can be computationally expensive. Exploring concatenating {s,a} and diffusing with one model could improve efficiency.**
>
> A1: Thank you for your insightful comments.
> We train two separate diffusion models since we adopt the diffusion-based generative behavior modeling method [1] as our backbone offline RL algorithm.
> This kind of backbone is well suited for continual learning problems, because training a unified behavior model can continually absorb new behavior patterns to promote forward knowledge transfer for the offline setting, and sampling from this generative model can naturally encompass a progressive range of observed behaviors.
> Then, for continual learning, we need to adopt another generative model to synthesize pseudo  samples of past tasks under the paradigm of deep generative replay [9, 10].
>
> On the other side, this architecture with two diffusion models could indeed increase computational costs.
> In literature, current diffusion-based RL methods [1-5] could face the challenge of increased computational costs due to the training and sampling of diffusion models.
> For example, the diffusion-planning methods [2, 6] also need to train two separate diffusion models, one to generate the trajectory data and another to predict the cumulative rewards of trajectory samples.
> We believe that the fast development of diffusion models in the generative modeling community can help tackle the above challenge for RL problems, such as sampling acceleration methods [7, 8].
> Our method is the first step that leverages expressive diffusion models to tackle the CORL challenge, and we will continue to tackle its limitations as stated in the original paper (bottom, page 9): "Though, our method requires two diffusion models to synthesize replayed samples, which could be further improved by sampling acceleration methods or developing one diffusion model for unifying the state and behavior modeling. We leave these directions as future work."
>
> **Q2: How does the continual training cost of updating diffusion models for each new task and sampling from them trade off with the memory savings of condensing previous tasks?**
>
> A2: In the continual learning setting, the category of generative replay methods [9, 10] needs to maintain two separate models: the solver (the backbone RL algorithm) and the generator (which mimics the data distribution of past tasks).
> In our method, we eliminate the dependency on accessing data of past tasks at the cost of maintaining an additional diffusion-based state generator.
> Taking the HalfCheetah task as an example, the state generator has about 24M parameters, and each new task has about 1M samples where each transition sample ($s,a,r,s'$) has a dimension of 26.
> As the number of tasks increases, the number of encountered samples will continue to expand, while the model size remains constant.
> Training the state generator with 1M samples roughly costs 2.5 hours on an NVIDIA RTX 3090 GPU, which could be further reduced to 50 minutes via distributed training on 4 GPUs.
> The advantage of generative replay becomes more remarkable for scenarios where massive datasets are needed to train a task or the continual learning setting contains a large number of sequential tasks.
> Additionally, the generative replay method can produce pseudo data as much as needed, which can also improve the robustness and flexibility of the model.
>
> Further, the pros of generative replay are not limited to memory savings only.
> In many practical applications, we cannot access real data due to privacy and security concerns, and generative replay is one of the few methods that is effective under these conditions.
> This challenge makes generative replay methods more necessary in real-world scenarios.
>
> [1] Huayu Chen, et al., Offline reinforcement learning via high-fidelity generative behavior modeling, ICLR 2023.
>
> [2] Michael Janner, et al., Planning with diffusion for flexible behavior synthesis, ICML 2022.
>
> [3] Zhendong Wang, et al., Diffusion Policies as an Expressive Policy Class for Offline Reinforcement Learning, ICLR 2023.
>
> [4] Bingyi Kang, et al., Efficient Diffusion Policies for Offline Reinforcement Learning, NeurIPS 2023.
>
> [5] Anurag Ajay, et al., Is Conditional Generative Modeling all you need for Decision Making? ICLR 2023.
>
> [6] Fei Ni, et al., MetaDiffuser: Diffusion Model as Conditional Planner for Offline Meta-RL, ICML 2023.
>
> [7] Cheng Lu, et al., DPM-Solver: A Fast ODE Solver for Diffusion Probabilistic Model Sampling in Around 10 Steps, NeurIPS 2022.
>
> [8] Qinsheng Zhang, et al., gDDIM: Generalized denoising diffusion implicit models, ICLR 2023.
>
> [9] Hanul Shin, et al., Continual learning with deep generative replay, NeurIPS 2017.
>
> [10] Rui Gao, et al., DDGR: Continual learning with deep diffusion-based generative replay, ICML 2023.

---

> ### Author Response · Authors · 2023-11-17
> **Response (Part 2/2)**
>
> **Q3: Comparison to previous continual RL algorithms in addition to diffusion-based model generator pipeline.**
>
> A3: As we stated in the paper, continual learning methods can be categorized into three kinds: regularization, parameter isolation (model expansion), and rehearsal.
> Each of these three types has its own pros and cons, which aligns with the No Free Lunch Theorem in machine learning research.
> For example, regularization methods can continually absorb new knowledge with a constant network capacity, while comprising additional regularization terms may lead to a tradeoff on the accomplishment of old and new tasks.
> Model expansion methods can leave existing knowledge totally undisturbed via parameter isolation, while it consumes increasing network resources.
>
> As we are the first to leverage expressive diffusion models to tackle the understudied CORL challenge, our primary concern is to validate whether our dual generative replay mechanism can realize high-fidelity replay of the sample space.
> Hence, following the classical DGR method [9], we compare the performance of the offline RL model trained with three variants of replay methods: Oracle, None, and Noise.
> For Oracle, the generated data is replaced with real data of past tasks to demonstrate that our diffusion-based generator can mimic the data distribution with high-fidelity.
> For Noise, the state generator is replaced with Gaussian noise to demonstrate the effectiveness of our diffusion-based state generator.
> For None, the state generator is removed and catastrophic forgetting is incurred, which demonstrates the necessity of our generative replay mechanism.
>
> We did not compare our method to several existing continual RL methods due to the difference in the intrinsic characteristics and applicable scenarios.
> For example, [11] belongs to the model expansion kind, [12] and [13] need to replay real samples of past tasks, and most of the existing methods [11, 12, 14] tackle the online RL problems other than the offline setting.
> On the other hand, it is significant to conduct empirical comparisons to these methods to demonstrate their respective pros and cons, or to modify these online continual methods to adapt to offline settings.
> We leave these for future work.
>
> [9] Hanul Shin, et al., Continual learning with deep generative replay, NeurIPS 2017.
>
> [11] Jean-Baptiste Gaya, et al., Building a subspace of policies for scalable continual learning, ICLR 2023.
>
> [12] Maciej Wolczyk, et al., Disentangling transfer in continual reinforcement learning, NeurIPS 2022.
>
> [13] Sibo Gai, et al., OER: Offline experience replay for continual offline reinforcement learning, ECAI 2023.
>
> [14] David Rolnick, et al., Experience replay for continual learning, NeurIPS 2019.

---

> > ### Comment · Reviewer_msqf · 2023-11-23
> > **Thanks for the response.**
> >
> > Thank the authors for the response. I’d like to keep my score for acceptance.

---

### Public Comment · ~Merriam_nikos1 · 2023-11-11
**The overall contribution is not clear within the writing and experiments**

The authors propose a practical paradigm that facilitates forward transfer and mitigates catastrophic forgetting to tackle sequential offline tasks with a discussion model and continual learning mechanism. But the following questions are terrible. The limited improved performance used so much computation. Is it the improved performance sourced from your proposed method not from the difussion and multi-head mechism? A+B+C does not equal to be a novel idea, it is rough. Why did you utilize the diffusion model?

---

> ### Author Response · Authors · 2023-11-18
> **Response**
>
> **Q1: Is it the improved performance sourced from your proposed method not from the diffusion and multi-head mechanism?**
>
> A1: As stated in our experiments, our primary concern is to validate whether our diffusion-based generative replay mechanism can realize high-fidelity replay of the sample space.
> Hence, we set three baselines as Oracle, None, and Noise.
> For Oracle, the generated data is replaced with real data from past tasks to demonstrate that our diffusion-based generator can mimic the data distribution with high fidelity.
> For Noise, the state generator is replaced with Gaussian noise to demonstrate the effectiveness of our diffusion-based state generator.
> For None, the state generator is removed and catastrophic forgetting is incurred, which demonstrates the necessity of our generative replay mechanism.
> In summary, these observations verify the contributions of our diffusion models to the continual learning performance.
>
> **Q2: Why did you utilize the diffusion model?**
>
> A2: A1: As repeatedly stated in our original paper, we have discussed the advantages of the diffusion model and the motivation for adopting it to tackle the CORL challenge.
> Due to the strong distributional expressivity and the ability to fit multimodal distributions, diffusion models have been explosively developed in the RL community recently [1-5], such as using diffusion models for behavior policy modeling [1], for target policy modeling [2], and for model-based trajectory modeling [2].
> In the first paragraph of Sec. 3.2 (bottom, page 4), we elaborate on the necessity of using the diffusion model to learn behavior from prior tasks as: "This perspective rooted in generative modeling presents three promising advantages for CORL. First, existing policy models are usually unimodal Gaussians with limited distributional expressivity, while collected behaviors in CORL become progressively diverse as novel datasets keep emerging. Learning a generative behavior model is considerably simpler since sampling from the behavior policy can naturally encompass a diverse range of observed behaviors, and allows the policy to inherit the distributional expressivity of diffusion models. Second, RL models are more prone to deficient generalization across diverse tasks. Learning a unified behavior model can naturally absorb novel behavior patterns, continually promoting knowledge transfer and generalization for offline RL. Third, generative behavior modeling can harness extensive offline datasets from a wide range of tasks with pretraining, serving as a foundation model to facilitate finetuning for any downstream tasks. This aligns with the paradigm of large language models, and we reserve this promising avenue for future research."
>
> On the other hand, other models such as behavior cloning, GANs, and VAEs can also be considered as alternatives to the diffusion model.
> In this paper, we choose the powerful diffusion model due to its promising properties discussed above.
> As the state-of-the-art architecture in the generative modeling community, the diffusion model has achieved great success in high-fidelity data generation, such as its superiority over GANs [6] and stable diffusion [7].
>
> [1] Huayu Chen, et al., Offline reinforcement learning via high-fidelity generative behavior modeling, ICLR 2023.
>
> [2] Zhendong Wang, et al., Diffusion Policies as an Expressive Policy Class for Offline Reinforcement Learning, ICLR 2023.
>
> [3] Michael Janner, et al., Planning with diffusion for flexible behavior synthesis, ICML 2022.
>
> [4] Bingyi Kang, et al., Efficient Diffusion Policies for Offline Reinforcement Learning, NeurIPS 2023.
>
> [5] Anurag Ajay, et al., Is Conditional Generative Modeling all you need for Decision Making? ICLR 2023.
>
> [6] Prafulla Dhariwal et al., Diffusion models beat GANs on image synthesis, NeurIPS 2021.
>
> [7] Robin Rombach et al., High-Resolution Image Synthesis with Latent Diffusion Models, CVPR 2022.

---

### Meta-Review · Area_Chair_JD5s · 2023-12-08

**Metareview:**

This submission proposes a paradigm for the simultaneous facilitation of forward transfer while mitigating catastrophic forgetting in the understudied context of sequential offline tasks. This is achieved through a dual generative replay framework in line with the popular approach of pseudo-reversal for Continual Learning. The dual model includes a generative process of states from past tasks as well as a behavior-generative model of actions given states. A critic network is specialized to specific tasks through multiple heads. State and behavior generative models are updated through interleaved samples from both models in combination with real samples for the current task, a standard method in pseudo-rehearsal approaches.

Strengths:
- First attempt to incorporate diffusion model for continual offline RL. Novel (msqf) Innovative (RUFG), technically sound (t1vB), promising (R6pv).
- Effectiveness in generating new samples and strong experimental results (msqf), empirical evidence from various tasks that demonstrates CuGRO's effectiveness (RUFG), empirical performance seems significant compared with considered baselines (t1vB)
- Good addition to an understudied problem (R6pv)

Weaknesses:
- Increase of computational cost due to training of two diffusion models (msqf, RUFG). This being the case for other models as well “generative replay methods needs to maintain two separate models” alone does not remove it as a weakness, especially since not all competing techniques need to store data instead (e.g. regularization methods).
- Lack of comparison to previous Continual RL algorithms (msqf, t1vB). Unfortunately “We did not compare our method to several existing continual RL methods due to the difference in the intrinsic characteristics and applicable scenarios […] We leave these for future work” is not a convincing counter-argument here
- Concerns about clarity (R6pv, t1vB)

Unfortunately, on balance, while being a promising approach in principle, a re-submission with a more thorough empirical evaluation will be needed for acceptance at ICLR or a similar venue.

**Justification For Why Not Higher Score:**

While this submission is promising in principle, too many concerns about empirical validity remain after the rebuttal. We strongly encourage the authors to resubmit once a more thorough empirical evaluation and comparison has been done.

**Justification For Why Not Lower Score:**

N/A

---

### Decision · Program_Chairs · 2024-01-16

Reject